# Investigating the Effects of Fairness Interventions Using Pointwise Representational Similarity

**Camila Kolling**                                    *ckolling@mpi-sws.org*
*MPI-SWS*

**Till Speicher**                                     *tspeicher@mpi-sws.org*
*MPI-SWS*

**Vedant Nanda**                                      *vnanda@mpi-sws.org*
*MPI-SWS*

**Mariya Toneva**                                     *mtoneva@mpi-sws.org*
*MPI-SWS*

**Krishna P. Gummadi**                                *gummadi@mpi-sws.org*
*MPI-SWS*

**Reviewed on OpenReview:** *https://openreview.net/forum?id=CkVlt2Qgdb*

## Abstract

Machine learning (ML) algorithms can often exhibit discriminatory behavior, negatively affecting certain populations across protected groups. To address this, numerous debiasing methods, and consequently evaluation measures, have been proposed. Current evaluation measures for debiasing methods suffer from two main limitations: (1) they primarily provide a *global* estimate of unfairness, failing to provide a more fine-grained analysis, and (2) they predominantly analyze the *model output* on a specific task, failing to generalize the findings to other tasks. In this work, we introduce Pointwise Normalized Kernel Alignment (PNKA), a pointwise representational similarity measure that addresses these limitations by measuring how debiasing measures affect the intermediate *representations* of *individuals*. On tabular data, the use of PNKA reveals previously unknown insights: while group fairness predominantly influences a small subset of the population, maintaining high representational similarity for the majority, individual fairness constraints uniformly impact representations across the entire population, altering nearly every data point. We show that by evaluating representations using PNKA, we can reliably predict the behavior of ML models trained on these representations. Moreover, applying PNKA to language embeddings shows that existing debiasing methods may not perform as intended, failing to remove biases from stereotypical words and sentences. Our findings suggest that current evaluation measures for debiasing methods are insufficient, highlighting the need for a deeper understanding of the effects of debiasing methods, and show how pointwise representational similarity metrics can help with fairness audits.

## 1 Introduction

Machine learning algorithms are now deeply integrated into several aspects of our daily lives. These algorithms not only recommend movies and products or suggest potential dating partners (Sun et al., 2019; Yu et al., 2023; Wu et al., 2022), but they are also increasingly employed in critical decision-making processes, such as approving loans and making hiring and health choices (Aletras et al., 2016; Liang et al., 2014; Sheikh et al., 2020). Despite their impressive performance, ML models face significant reliability challenges, particularly in decision-making tasks (Geirhos et al., 2018; Hendrycks & Dietterich, 2019; Taori et al., 2020;

Szegedy et al., 2013; Papernot et al., 2016). A major concern is their discriminatory behavior against certain protected groups (Angwin et al., 2016; O'neil, 2017; Buolamwini & Gebru, 2018), manifesting as biases that adversely affect individuals based on race, gender, age, or other protected characteristics. These biases can result in unfair outcomes that negatively affect individuals based on their characteristics. To mitigate such discriminatory behavior, researchers have proposed a variety of approaches that intervene at various stages of the ML pipeline, including data pre-processing (Feldman et al., 2015; Quadrianto et al., 2019; Ryu et al., 2018; Wang & Deng, 2020), learning fair intermediate representations (Dwork et al., 2012; Zemel et al., 2013; Edwards & Storkey, 2016; Madras et al., 2018; Beutel et al., 2017; Wang et al., 2019) [1], in-processing by adding constraints to the objective function (Zafar et al., 2017b;a; 2019; Donini et al., 2018; Goel et al., 2018; Padala & Gujar, 2020; Agarwal et al., 2018; Martinez et al., 2020; Diana et al., 2020; Lahoti et al., 2020), and post-processing by changing model outputs (Hardt et al., 2016; Wang et al., 2020; Savani et al., 2020).

Following the introduction of these debiasing methods, the research community has developed several evaluation measures to assess their effectiveness. These measures aim to quantify the fairness of ML models and include metrics such as equalized odds, equality of opportunity, and demographic parity Hardt et al. (2016); Dwork et al. (2012); Kusner et al. (2017). However, current evaluation approaches face two main limitations. First, they primarily focus on providing a *global estimate* of unfairness by analyzing the disparities between groups categorized by a single protected attribute, such as race or gender (Hardt et al., 2016; Dwork et al., 2012; Kusner et al., 2017). However, as recent work reveals (Buolamwini & Gebru, 2018; Kearns et al., 2018; Kim et al., 2020), fairness is multifaceted and can manifest in several different ways, making a more fine-grained analysis necessary. Second, fairness evaluation approaches predominantly analyze model behavior on a specific target task (Hardt et al., 2016; Dwork et al., 2012; Kusner et al., 2017). In most cases they do not consider the *representations* that models use to reason about the data. Conceptually, most algorithms first compute an intermediate representation $z = g(x)$ of an input $x$, before deriving an outcome $y = f(z)$. Such intermediate representations appear, for example, as a result of feature engineering or at the intermediate layers of deep neural networks, and are becoming increasingly important with the rise of large, pretrained *foundation models* (Bommasani et al., 2021; Radford et al., 2021; Brown et al., 2020), where the representations of one model serve as the basis for many different downstream tasks. The insights from prior methods that only focus on task-specific outcomes $y$ cannot easily be generalized to other tasks that use the same representations $z$.

In this work, rather than only evaluating task-specific outcomes on a global level, we address the limitations of previous evaluation measures by analyzing *how debiasing methods modify the intermediate representations of individual data points*. More specifically, we analyze how much the representation of a data point changes from "baseline" (non-debiased) to "fair" (debiased) models. To do so, we propose a new measure that is inspired by the existing rich body of research on representational similarity measures in the machine learning community (Laakso & Cottrell, 2000; Li et al., 2015; Wang et al., 2018; Raghu et al., 2017; Morcos et al., 2018; Kornblith et al., 2019). Representation similarity measures, such as the widely-used CKA (Kornblith et al., 2019), analyze how two different models represent a given dataset. They capture similarity as a real number that reflects the degree of similarity between the representations, with 1 indicating identical representations. These measures thus allow us to analyze how model interventions impact all downstream tasks that use these representations.

An important limitation of previously introduced representation similarity measures is that they only provide an aggregate similarity score across the entire dataset. However, such aggregate scores are not suitable for achieving our goal of analyzing at a local level how debiasing methods are affecting *individuals*. To enable the fine-grained study of representation similarity, we modify the widely-used CKA measure (Kornblith et al., 2019) into a *pointwise representational similarity measure*, which we call Pointwise Normalized Kernel Alignment (PNKA). PNKA provides a similarity score for each individual data point, allowing us to study how model interventions affect individuals at the representation level.

Our key contributions are summarized as follows:

---

[1] Prior work has included learning fair representation in both pre- and in- processing, e.g., see Zafar et al. (2019); here we choose to list it separately since it is the main focus of our paper.

- We introduce PNKA, a pointwise representation similarity measure. PNKA allows us to analyze the effect of model interventions, such as debiasing, at a local level, by measuring how interventions change the representations of individuals. PNKA is broadly applicable to all debiasing approaches that modify the data or the models. Moreover, it offers a more generalized solution by eliminating the need for manually developing domain-specific evaluation methods for each individual use case.

- We demonstrate PNKA's utility in auditing representations of debiasing approaches. On the COM-PAS and Adult datasets, PNKA reveals previously unknown insights: while group fairness predominantly influences a small subset of the population, maintaining high representational similarity for the majority, individual fairness constraints uniformly impact representations across the entire population, altering nearly every data point. Our observations on representation similarity allow us to predict the effect that training ML models on debiased representations will have, and we demonstrate that the actual outcomes do indeed match the predictions.

- By applying PNKA to contextual and non-contextual language embeddings, we show that debiasing methods in these domains may not perform as intended. Specifically, PNKA shows that debiasing approaches do not consistently remove gender properties from stereotypical words and sentences as anticipated. Our results also suggest that the fairness evaluation measures currently employed for evaluating (non-)contextual debiased embeddings are both limited and insufficient for comprehensively assessing these debiasing methods.

## 2 Pointwise Representational Similarity Measure for Assessing Debiasing Approaches

When an algorithm processes data $x$ for an individual, it typically operates in two stages: 1) computing an *(intermediate) representation $z = g(x)$* and 2) computing the *outcome $y = f(z)$*. To ensure that algorithms produce fair outcomes, a popular approach is to replace a potentially biased feature extractor $g$ with an unbiased version $g'$, such that the modified representation $z' = g'(x)$ does not contain information about a sensitive feature (Zemel et al., 2013; Lahoti et al., 2019). We use the notation $z$ here to refer to the representation of a single individual $x$, where $z$ is a $d$-dimensional vector, i.e., $z \in \mathbb{R}^d$. We use uppercase $Z$ to refer to the representations of a set of $N$ individuals, i.e., $Z \in \mathbb{R}^{N \times d}$.

When debiasing feature extractor $g$, it is important to be able to assess the effects of the change on performance and fairness. Most prior work focuses on studying how changes to $g$ affect the outcomes $y$ (Hardt et al., 2016; Dwork et al., 2012; Kusner et al., 2017), but the insights from such assessments only apply to specific downstream tasks. To gain a better understanding of how changes to the representations $Z$ will impact different downstream tasks, we have to study $Z$ directly. Such an assessment can be performed using a *representation similarity measure $s(Z, Z')$* that measures how similar $Z'$ is to $Z$. Representational similarity measures typically provide a single overall score that estimates how similar two different representations of an entire set of input points are. However, overall representation similarity scores do not allow us to assess how much the representation for *individual points* change, and can thus overlook potential adverse effects that changes to the representation can have on small groups or individuals.

To address these issues, we adopt a *pointwise* measure that assigns an individual representational similarity score to each data point. With this measure, we can effectively determine whether data or model interventions, such as debiasing, impact all instances uniformly (i.e., whether all individuals are affected by the debiasing method) or disproportionately impact certain data points (i.e., some individuals are more affected than others). By comparing representations of a baseline (non-debiased) model $g$ with its debiased version $g'$, we can identify and characterize which individuals are most affected and whether fairness interventions effectively target those they are aimed at. For instance, by focussing on the individuals with the lowest similarity scores, we can determine whose representation change the most from the baseline to the debiased version. Next, we describe our pointwise representation similarity measure.

## 2.1 Intuition for Pointwise Similarity Across Representations

An initially appealing way to measure representational similarity of the $i$-th point in representations $Z$ and $Z'$ is to directly apply a (dis)similarity metric, such as the Euclidean distance or cosine similarity, to its two representations $Z_i$ and $Z_i'$, e.g., by defining $s(Z, Z', i) = \cos(Z_i, Z_i')$. One immediate failure mode of such an approach is when $Z_i$ and $Z_i'$ have a different number of dimensions $d \neq d'$. However, even when the number of dimensions matches, any such approach that directly compares the two representations suffers from a subtle but important shortcoming of not being invariant to orthogonal transformations. Consider an example where $Z = R Z'$ with $R$ being an orthogonal matrix, such that $Z_i^\top Z_i' = 0 \ \forall i$. Even though $cos(Z_i, Z_i') = 0 \ \forall i$, i.e., representations appear very dissimilar when directly measuring their cosine similarity, they are, however, from an information-theoretic standpoint, identical for any downstream applications[2]. Therefore, the low similarity score obtained from directly comparing points is misleading. In fact, previous work LeCun et al. (1990); Xie et al. (2017); Liu et al. (2021) has shown that orthogonal transformations do not change the training dynamics of neural networks and thus invariance to them is a desirable property for any similarity metric operating on neural representations, as also discussed in Kornblith et al. (2019). A similar argument could be made against other choices of direct comparisons such as Euclidean distance.

To overcome the issues involved in directly comparing two different representations, we propose an indirect comparison. We leverage the simple, but powerful insight from prior work Kornblith et al. (2019); Kriegeskorte et al. (2008) that while we cannot directly compare similarity *across* representations, we can do so *within* the same representation. We argue that the *representations $Z_i$ and $Z_i'$ of a point $i$ should be considered similar across representations $Z$ and $Z'$ if their positions relative to other points in the respective representation are similar*. Therefore, to determine whether the representations $Z_i$ and $Z_i'$ of point $i$ are similar, we can first compare how similarly $i$ is positioned relative to all the other points within each representation. We then compare the relative position of $i$ across both representations. This approach builds on well-established principles from both representation similarity literature Kriegeskorte et al. (2008); Gower (1975); Kornblith et al. (2019) and neuroscience Kriegeskorte & Kievit (2013); Cichy et al. (2017); Mehrer et al. (2021), where shifts in relative positioning are key to analyzing changes in (brain) representations. Prior work, such as CKA, discusses the ongoing debate on the desirable properties of robust representation similarity measures, and we adopt the properties it formulates as the basis for our analysis.

## 2.2 Measuring Similarity in the Relative Position of Points

We can now formally describe our proposed measure, Pointwise Normalized Kernel Alignment (PNKA), which calculates representational similarity for individual points by first comparing the relative similarity between a point and other points within the same representation, and then across the two different representations. Given a set of (column-centered) representations $Z$ (and analogously for $Z'$) and a kernel $k(.,.)$, we can define a pairwise similarity matrix between all $N$ points in $Z$ as $K(Z)$ with $K(Z)_{i,j} = k(Z_i, Z_j)$. In our work, we use linear kernels, i.e., $k(Z_i, Z_j) = Z_i^\top \cdot Z_j$, but other kernels, e.g., RBF Kornblith et al. (2019) kernels, could be used as well [3]. We leave the exploration of other types of kernels for future work. Given two similarity matrices $K(Z)$ and $K(Z')$, we measure how similarly point $i$ is represented in $Z$ and $Z'$ by comparing its position relative to all other points. To this end, we define

$$\text{PNKA}(Z, Z', i) = \cos(K(Z)_i, K(Z')_i) = \frac{K(Z)_i^\top K(Z')_i}{||K(Z)_i|| \ ||K(Z')_i||}, \tag{1}$$

where $K(Z)_i$ and $K(Z')_i$ denote how similar point $i$ is to all other points in $Z$ and $Z'$, respectively. We use cosine similarity to compare the within-representation similarity of a point with the other points, across representations, for two reasons. First, cosine similarity provides us with normalized similarity scores for each point. Second, by normalizing by the length of the similarity vectors $K(Z)_i$ and $K(Z')_j$, we compare the *relative* instead of the absolute similarity of points, i.e., how similar point $i$ is represented relative to points $j$ and $j'$. We can further extend our measure into an aggregate version ($\overline{\text{PNKA}}$) between sets of representations by computing the average of the similarities across all the $N$ points.

---

[2]The projection matrix $R$ is invertible, so by multiplying the weight matrix of any linear downstream operation on $Z$ with $R^{-1}$, it can be directly applied to $Z'$ and produce the same result.

[3]Results using the RBF kernel are consistent with those from the linear kernel, leading to similar conclusions. Detailed results for all evaluation sections are provided in the corresponding appendices.

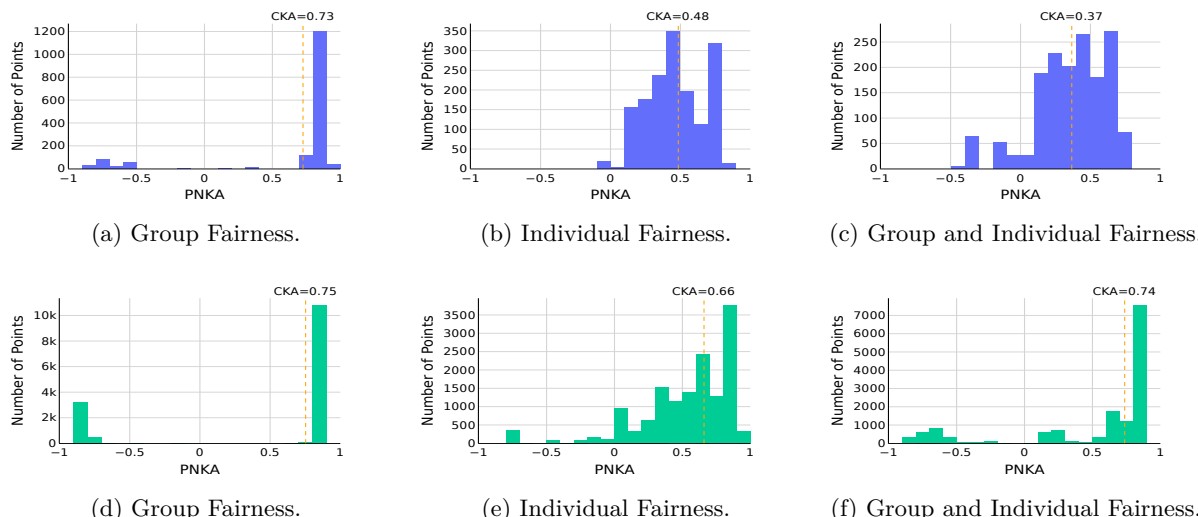

Figure 1: Distribution of PNKA similarity scores. The first row (blue plots) shows results for the COMPAS dataset, debiased with respect to race, while the second row (green plots) displays results for the Adult dataset, debiased based on gender. The vertical dotted line shows the overall similarity scores provided by CKA Kornblith et al. (2019). We compare the baseline representations (trained only for utility) with those from models trained using three different loss functions: Utility + Group Fairness (a, d), Utility + Individual Fairness (b, e), and Utility + Group and Individual Fairness (c, f).

## 2.3 Properties of PNKA

**PNKA captures the similarity between neighborhoods.** Previously we argued that for a data point to be similarly represented across two representations, it should be similarly positioned relative to the other points in each representation. In Appendix A.1 we empirically show, across several architectures, datasets, distances measures, and values of $k$, a clear relationship between high PNKA scores and high overlap of nearest neighbors across representations, indicating that PNKA captures how similar the neighborhoods of the points are. We further show in the same Appendix that the direct comparison of representations through cosine similarity does not exhibit the same trend.

**Relationship of PNKA with aggregate measures of representational similarity.** PNKA is inspired by CKA, and in Appendix A.2 we empirically show that the aggregate version of our measure ($\overline{\text{PNKA}}$) produces results close to CKA.

**Invariance properties.** Previous work Kornblith et al. (2019) has identified invariance to orthogonal transformations and isotropic scaling as desirable properties for representational similarity measures. We provide a mathematical proof that our measure possesses both of these invariance properties in Appendix A.3.

## 3 Investigating Debiased Tabular Data Representations

Prior work (Zemel et al., 2013; Creager et al., 2019; Louizos et al., 2015; Xu et al., 2018) has proposed learning fair data representations which retain minimal information about sensitive features. Here, we conduct a case study to verify that PNKA works as expected when auditing the effect of debiasing approaches. Our goal in this section is to establish that PNKA yields meaningful and reliable insights.

To achieve our goal, we compare the original (baseline) and debiased representations, focusing on well-established debiasing methods. One such techniques is the approach proposed by Zemel et al. (2013), which learns representations of tabular data by optimizing a loss function that maintains as much utility as possible, while removing information about protected attributes. The learning algorithm modifies the data representation based on three distinct objectives: classification accuracy (denoted by us as utility), statistical

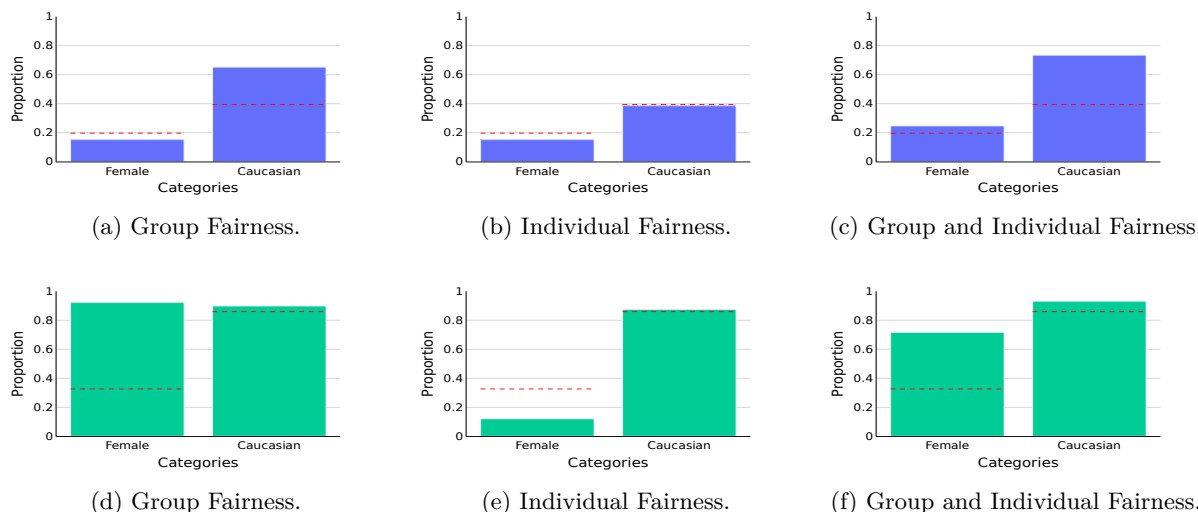

Figure 2: Distribution of the binary attributes for the 10% most affected individuals (i.e., lowest PNKA score). The first row (blue plots) shows results for the COMPAS dataset, debiased with respect to race, while the second row (green plots) displays results for the Adult dataset, debiased based on gender. We compare the baseline representations (trained only for utility) with those from models trained using three different loss functions: Utility + Group Fairness (a, d), Utility + Individual Fairness (b, e), and Utility + Group and Individual Fairness (c, f). The horizontal red dotted line shows the population average per attribute.

parity (to achieve group fairness, and data loss (as a proxy for achieving individual fairness). By applying combinations of these objectives, we obtain four types of representations, one based on utility, serving as the baseline, and three debiased versions: utility and group fairness, utility and individual fairness, and a combination of all three objectives (utility, group and individual fairness).

We use the COMPAS (Larson et al., 2016) dataset, debiased for race, and the Adult (Becker & Kohavi, 1996) dataset, debiased for gender [4], to analyze with PNKA the effect of the different debiasing objectives. We first investigate the overall effect of debiasing in Section 3.1, followed by a more detailed analysis of the individuals whose representations change the most in Section 3.2. Finally, we show that the predictions made based on PNKA scores match the predictions observed on downstream tasks in Section 3.3.

## 3.1 How Do Individual Representations Change?

The distribution of PNKA similarity scores, obtained by comparing the representations of the baseline with each of the debiasing methods, is visualized in Figure 1, with COMPAS Larson et al. (2016) results depicted in the first row (Figures 1a– 1c), and Adult Becker & Kohavi (1996) shown in the second row (Figures 1d– 1f). As shown in Figures 1a and 1d, for both datasets, under group fairness, the majority of individuals maintain high representational similarity, suggesting that the majority of individuals' representations remain similar to the baseline, with significant changes primarily occurring for a small subset of the population. In contrast, as depicted in Figures 1b and 1e, individual fairness constraints lead to more uniform representational change across the entire population, slightly impacting nearly every data point's representation. Finally, when combining group fairness and individual fairness, as observed in Figures 1c and 1f, the impact on the data points' representations varies depending on the dataset. For the COMPAS dataset, the resulting pattern closely resembles that observed under the influence of only group fairness. In contrast, the pattern observed for the Adult dataset aligns more closely with the effects seen when applying only individual fairness.

---

[4]We pre-process each dataset as done by Zemel et al. (2013). More information can be obtained in `https://github.com/Trusted-AI/AIF360`.

## 3.2 Whose Representations Change the Most?

Next, we analyze the 10% of the population with the lowest PNKA similarity scores, representing those whose representations have undergone the most significant changes due to the applied fairness constraints. We focus on the distribution of the protected attributes in this group, specifically on race and gender. Figure 2 illustrates the distribution of the binary sensitive attributes for the subset of the most affected people in both COMPAS, in the first row, and Adult, in the second row. The horizontal red dotted line shows the population average per attribute [5].

In the COMPAS dataset (Figures 2a– 2c), where data representations are debiased according to race, we observe that the gender distribution remains relatively stable, aligning closely with the population baseline distribution, while the race distribution exhibits notable shifts. More specifically, for COMPAS, under the group fairness objective, the individuals whose representations are altered the most are primarily Caucasians. This happens regardless of whether the group fairness constraint has been used alone or in combination with individual fairness. However, when only individual fairness is applied, the distribution resembles that of the baseline one, suggesting a minimal impact of individual fairness on attribute distribution compared to group fairness.

In contrast, as shown in Figures 2d–2f, for the Adult dataset, whose representations are debiased with respect to gender, under group fairness conditions, the female category is the most drastically affected group. This happens regardless of whether individual fairness is applied in combination with group fairness. Interestingly, however, when individual fairness constraints are applied, the male representations are more affected than female ones. The race attribute largely retains its baseline distribution.

## 3.3 Do PNKA's Predictions Match Downstream Outcomes?

The previous results suggest that under some debiasing objectives, the data representations of specific demographics are disproportionally affected. However, do the changes in the data representations also translate to changes in the behavior of models that use them? In other words, can the analysis at the representation level provide insights at the output level?

To test the utility of PNKA in predicting downstream behavior, we use the insights obtained from PNKA in Sections 3.1 and 3.2, together with prior knowledge about the design of the fairness constraints, to predict what outcomes we can expect from models that use the modified data representations. We then train models on the representations to test whether the predictions made using PNKA match the actual outcomes. In our analysis, we focus on the effect of the GF-constraints, because (i) we observe in Section 3.1 that they strongly affect a small part of the population that we can easily characterize in Section 3.2 (as opposed to the more uniform changes across the board for the other constraints, which make it harder to characterize the demographics of the affected individuals), and (ii) it is easy to understand their effect, namely minimizing disparities between groups that differ in the protected attribute.

**Hypothesis:** For the COMPAS dataset, we observe in Figure 2a that the people whose representations are changed the most by the group fairness constraints are predominantly Caucasians. Given that the COMPAS dataset is known to exhibit a bias in favor of Caucasians, i.e., that they are less likely to recidivate than non-Caucasians, a change in the representations of Caucasians therefore suggests that a model using the group fairness debiased representations might assign Caucasians less favorable outcomes than one that uses the baseline representations. Conversely, for the Adult dataset we observe in Figure 2d that females are the predominant demographic in the 10% of most affected representations. Given that in this dataset males tend to be more likely to have the positive label (income above 50K per year), we thus expect that a model using group fairness debiased representations will assign more favorable outcomes to females compared to a model using baseline representations.

To test whether these hypotheses, based on PNKA's similarity scores, are correct, we train logistic regression models for each of the representation types. On the COMPAS dataset the goal is to predict recidivism, whereas on the Adult datasets the goal is to predict whether an individual's income is above 50K dollars

---

[5]We show the distribution of the remaining attributes in Appendix B.2.

per year. The overall accuracy and debiasing results of these models are presented in Table 1. Following the approach used by Zemel et al. (2013), we measure group fairness through the statistical parity difference, which is the ratio of favorable outcomes received by unprivileged versus privileged classes. In the context of the COMPAS dataset, where the model predicts the risk of criminal recidivism, a positive (negative) score indicates a higher predicted risk for non-Caucasians (Caucasians). For the Adult dataset, where the model is trained to predict whether income exceeds 50K per year, a positive (negative) score means females (males) are less likely to receive a high annual income. Individual fairness (Zemel et al., 2013) is assessed using the consistency score, which evaluates how similar the predicted labels are for neighboring instances. Table 1 shows that in both the COMPAS and Adult datasets, the inclusion of group fairness leads to models exhibiting a lower statistical parity difference compared to the baseline. This suggests a reduction in bias against non-Caucasians for COMPAS, and females for the Adult dataset.

To better understand whether the reduced statistical parity on both datasets is indeed due to the hypothesized changes for each of the groups, we investigate how the benefits that each of the groups receives on the two datasets changes. In the case of COMPAS, false positives are deemed to be far more costly – the widely accepted Blackstone's ratio posits that "it is better that ten guilty persons escape than that one innocent suffer" (Blackstone, 2016). On the other hand, in the case of Adult, false negatives are deemed to be more costly, as it suggests that the individual has a lower income, which can be associated with limited financial resources, reduced access to opportunities, and potentially lower socio-economic status. Thus, for COMPAS, we measure precision and for the Adult dataset, recall, since these measures capture the change in beneficial outcomes.

The results for each protected group are shown in Table 2. The results corroborate the earlier predictions made by PNKA. For the COMPAS dataset, the use of group fairness significantly impacts Caucasians, leading to a notable decrease in their precision score. In particular, the precision score changes more strongly for Caucasians ($\sim 13\%$) than for non-Caucasians ($\sim 1\%$). On the Adult dataset, the use of group fairness results in a substantial increase of recall scores for females. Again, females experience a much larger change in recall ($\sim 34\%$) than males ($\sim 6\%$). Both of these findings match the predictions made using PNKA, i.e., the groups achieving the largest change in their outcomes are those that are most prevalent among the individuals whose representations changed the most due to the group fairness constraints.

***Takeaways.*** The results in this section show that PNKA can be a valuable auditing tool, as it enables a comprehensive and detailed analysis of the changes that occur when modifying representations. Further, we demonstrate that the insights provided by PNKA are reliable, exhibiting predictive power for downstream applications and aligning with findings from prior outcome-based studies. Importantly, unlike prior work, which requires studying each downstream application of the representations separately and depends on prior knowledge to identify relevant groups, PNKA directly quantifies the extent of change in representations at the individual level, offering a more targeted and efficient approach.

| Dataset | Constraint Type | Accuracy | Statistical Parity | Consistency |
|---------|----------------|----------|--------------------|-------------|
| COMPAS | Utility | 0.6780 | 0.2308 | 0.9811 |
| | Utility + Group Fairness | 0.6585 | 0.0376 | 0.9220 |
| Adult | Utility | 0.8015 | -0.1784 | 0.9385 |
| | Utility + Group Fairness | 0.7878 | -0.0426 | 0.9968 |

Table 1: Overall accuracy, statistical parity, and consistency results for linear regression models trained with baseline (Utility) data representation and with the group fairness debiased representations. For statistical parity, in the context of the COMPAS dataset, where the model predicts the risk of criminal recidivism, a positive score indicates a higher predicted risk for non-Caucasians. On the other hand, in the Adult dataset, where the model is trained to predict whether income exceeds 50K per year, a negative score means fewer females are likely to receive a high annual income. Ideally, statistical parity should be 0, while a score of 1 is optimal for both accuracy and consistency.

| Measure | Dataset | Protected Attribute | Data | |
|---|---|---|---|---|
| | | | Utility | Utility + Group Fairness |
| Precision | COMPAS | Caucasian | 0.6797 | 0.5465 |
| | | Non-Caucasian | 0.6980 | 0.7090 |
| Recall | Adult | Male | 0.4340 | 0.3731 |
| | | Female | 0.0752 | 0.4146 |

Table 2: Precision and recall scores for COMPAS and Adult datasets, respectively, of linear regression models trained with baseline (U) data representation and the group fairness debiased data representations. In the COMPAS dataset, where there is a known bias against non-Caucasians, we found that precision for Caucasians is reduced due to group fairness interventions, indicating an adverse effect. In the Adult dataset, characterized by a bias against females, females are positively affected by group fairness.

## 4 Investigating Debiased Language Embeddings

In our next case study, we analyze PNKA as a tool for investigating debiased word embeddings. Previous work Bolukbasi et al. (2016); Gonen & Goldberg (2019); Lu et al. (2020) has identified the presence of stereotypical biases in word embeddings, i.e., vector representations of words, and since has developed several debiasing methods Bolukbasi et al. (2016); Zhao et al. (2018); Kaneko & Bollegala (2019). Among them are: (1) Gender Neutral (GN-)GloVe Zhao et al. (2018), which focuses on disentangling and isolating all the gender information into certain specific dimension(s) of the word vectors; and (2) Gender Preserving (GP-)GloVe Kaneko & Bollegala (2019), which targets preserving non-discriminative gender-related information while removing stereotypical discriminative gender biases from pre-trained word embeddings. The latter method is also used to finetune GN-GloVe embeddings, creating a third, combined method called GP-GN-GloVe.

We start, in Section 4.1, by briefly explaining how these debiasing approaches are traditionally evaluated. In Section 4.2, we show how PNKA can be leveraged to investigate whether these approaches modify the group of words (i.e., stereotypical words) as originally intended. Using the insights from PNKA, we formulate hypothesis about the effects of the debiasing approaches, and test them in Section 4.3.

### 4.1 Traditional Measures for Debiased Word Embeddings

The aforementioned debiasing models for word embeddings have been originally evaluated using Sem-Bias Zhao et al. (2018), a dataset designed to assess whether the debiasing methods have successfully removed stereotypical gender information from the word embeddings. Each instance in SemBias consists of four word pairs: a *gender-definition* word pair (e.g. "waiter - waitress"), a *gender-stereotype* word pair (e.g. "doctor - nurse"), and two other word-pairs that have similar meanings but no gender relation, named *gender-neutral* (e.g. "dog - cat"). To assess the bias in word embeddings, the evaluation scheme measures the cosine similarity with the canonical gender vector, i.e., $cos(\overrightarrow{a} - \overrightarrow{b}, \overrightarrow{he} - \overrightarrow{she})$ for each of the four word pairs $(a, b)$ in a SemBias instance. The word pair with the highest cosine similarity is selected as the "predicted" answer. If the word embeddings are correctly debiased, then the cosine similarity of the $\overrightarrow{he} - \overrightarrow{she}$ vector with the gender-definition words should be high, and the similarity with the gender-stereotype words should be low, i.e., the frequency of predictions for these categories should be high for the gender-defining word pairs and low for gender stereotypical word pairs.

Thus, GP- and GN-GloVe evaluate how (de)biased embeddings are based on whether they predict stereotypical or definitional word pairs in each instance of the SemBias dataset. We show results for evaluating GP- and GN-GloVe on SemBias in Appendix C.2. The evaluation shows that GP-Golve embeddings offer only a marginal improvement over the baseline embeddings, while GN-GloVe and GP-GN-GloVe embeddings show substantial reductions in bias in the prediction task. These findings suggest that the latter models are more effective in mitigating gender bias in word embeddings.

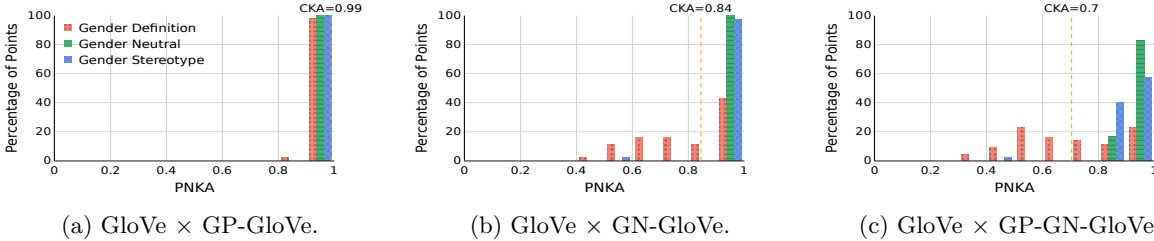

(a) GloVe × GP-GloVe.  (b) GloVe × GN-GloVe.  (c) GloVe × GP-GN-GloVe.

Figure 3: Distribution of PNKA scores per group of words for SemBias dataset Zhao et al. (2018). We compare the baseline (GloVe) model and its debiased versions. Words with the lowest similarity scores are the ones that change the most from the baseline to its debiased version. Across all debiased embeddings, the words whose embeddings change the most are the gender-definition words.

## 4.2 How Do Individual Representations Change?

We next employ PNKA to better understand which word embeddings have changed the most due to the debiasing procedure of GP- and GN-GloVe methods. As before, we use PNKA to measure similarity between the original GloVe (baseline) and the debiased versions of GloVe embeddings, i.e., (GN-)GloVe Zhao et al. (2018), Gender Preserving (GP-)GloVe and GP-GN-GloVe Kaneko & Bollegala (2019). Figure 3 shows the distribution of PNKA similarity scores for words in the SemBias dataset grouped by their category (i.e., gender defining, gender neutral, and gender stereotype).

We first observe, in Figure 3a, that GP-GloVe representations exhibit a high degree of similarity to the original GloVe embeddings for almost all words. This suggests that the GP-GloVe method may not significantly alter the word representations, maintaining a close resemblance to the original embeddings for most words. However, Figures 3b and 3c show that GN-GloVe and GP-GN-GloVe, respectively, considerably change the representations for a subset of the words. This observation aligns well with the results of the prior evaluation, detailed in Table 4 of Appendix C.2, in which GP-GloVe was shown to yield results similar to GloVe (suggesting similar representations), while GN-GloVe and GP-GN-GloVe achieve better debiasing results.

Moreover, as illustrated in Figure 3, an intriguing pattern emerges across all three debiasing methods: the words with lower PNKA scores, i.e. the words whose representations change the most, belong predominantly to the gender-definition category. This finding stands in stark contrast to the expected behavior: As discussed in Zhao et al. (2018) and Kaneko & Bollegala (2019), the primary goal of these debiasing techniques for word embeddings is to retain gender-specific information in feminine and masculine words (i.e., gender-defining words), maintain neutrality in gender-neutral words, and eliminate biases in stereotypical words. Thus, the expectation is that debiasing should primarily alter gender-stereotypical word embeddings, while mostly preserving gender-definitional ones.

## 4.3 Do PNKA's Predictions Match the Projection Analysis?

Our analysis using PNKA reveals that, contrary to the expected effect of debiasing, the most profound changes occur in the gender-defining words, challenging the conventional understanding of how these debiasing methods function and prompting a more careful analysis of their effects. We formulate a new hypothesis about the impact of debiasing on word embeddings: *instead of removing the gender information in gender-stereotypical words as initially intended, debiasing methods inadvertently amplify gender information in the gender-definition words.* Such an effect could be missed by the conventional evaluation procedure discussed previously, which only assesses the *relative* cosine-similarity. Increasing the gender-related information in gender-defining words would make gender-defining words more gender-aligned, and thus increase their prediction frequency, relative to other word pairs in SemBias. This effect could be achieved, however, without removing gender-related information from the gender stereotypical words, as intended by the debiasing methods.

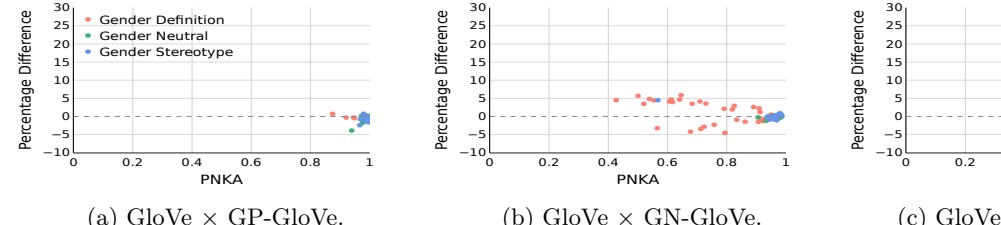

(a) GloVe × GP-GloVe.  (b) GloVe × GN-GloVe.  (c) GloVe × GP-GN-GloVe.

Figure 4: Relationship between PNKA scores (x-axis) and percentage difference (y-axis) in magnitude of the projection on the gender direction $\overrightarrow{he}$ - $\overrightarrow{she}$. A positive or negative percentage difference value indicates a shift in magnitude along the gender direction. Word embeddings that change their gender information are the ones that obtain low PNKA scores.

To test this hypothesis, we measure for each word how much its embedding changed in terms of gender information, when compared to the original GloVe embedding, by projecting it onto the canonical gender vector $\overrightarrow{he}$ - $\overrightarrow{she}$. More specifically, for each embedding approach $e$ and word $i$, we project the corresponding word embeddings $\phi_i^{(e)} = e(i)$ onto the gender vector direction $g^{(e)} = \overrightarrow{\phi_{he}^{(e)}}$ - $\overrightarrow{\phi_{she}^{(e)}}$ and compute the projection magnitudes $p_i^{(e)} = \phi_i^{(e)} \cdot \hat{g}^{(e)\top}$, where $\cdot$ represents a dot product, and $\hat{g}$ is the normalized gender direction. The higher $p_i^{(e)}$ is, the more gender information is contained in the word embedding vector $\phi_i^{(e)}$. To understand how much each of the debiased embedding methods change the amount of gender information, relative to the original *glove* embeddings, we analyze the percentage difference in magnitude, defined as $\omega_i^{(e)} = \frac{p_i^{(e)} - p_i^{(glove)}}{|p_i^{(glove)}|}$. A $\omega_i^{(e)} = 0$ indicates that the gender information in the debiased embedding has not changed relative to (baseline) GloVe, while $\omega_i^{(e)} > 0$ ($\omega_i^{(e)} < 0$) indicates an increase (decrease) in the male (female) gender information associated with word $i$.

Figure 4 depicts the relation between PNKA scores (x-axis) and the percentage difference in magnitude (y-axis) for each word in the SemBias dataset. We can see that the GN-GloVe and GP-GN-GloVe debiasing methods primarily amplify the gender information in gender-definition words (red dots), rather than reduce it for gender-stereotype words (blue dots), i.e., $\omega_i^{(e)} \neq 0$ for gender-defining and $\omega_i^{(e)} \approx 0$ for gender-stereotype words. The words exhibiting the most significant change in gender information, identified as the ones with lower PNKA similarity scores, predominantly fall within the gender-defining category. This observation supports our alternative explanation that these debiasing methods might be enhancing gender information in gender-defining words, rather than diminishing it in gender-stereotypical words.

Finally, we applied a similar analysis using PNKA to investigate bias mitigation efforts in contextualized embeddings of transformer-based language models. Unlike static word embeddings, these models generate word representations that depend on surrounding context, which makes identifying and mitigating biases challenging yet crucial, given that biases in the pre-trained models can propagate to numerous downstream tasks. We focused on a debiasing method that aims to remove gender bias from contextual representations of stereotypical words (e.g., associating "math" with a male bias or "poetry" with a female bias) while retaining gender-specific information in gender-defining words. Our analysis using PNKA reveals that, similar to findings with static embeddings, the debiased contextualized embeddings showed only minimal shifts along the gender direction, failing to reduce bias in the intended stereotypical contexts. Detailed results from this investigation are available in Appendix D.

***Takeaway.*** The findings in this section shows that PNKA is a powerful tool for auditing word embedding debiasing techniques, as it allows for a detailed, representation-level analysis of how gender information is altered by these methods. Through PNKA, we gain reliable insights that not only complement traditional outcome-based evaluations but also reveal underlying representational shifts that might otherwise go unnoticed. Crucially, unlike prior approaches that rely on predictive task assessments or predefined gender categories, PNKA quantifies representational changes at the individual word level, enabling a more precise

and efficient evaluation of bias mitigation efforts. This ability to identify specific shifts in gender-related representations positions PNKA as an essential tool in refining and improving debiasing strategies.

## 5 Related Work

In this section, we first review prior work on developing and assessing debiasing methods in the fairness domain. We then review work on representational similarity measures.

### 5.1 Fairness in ML Systems

ML algorithms have become fundamental to several aspects of daily life, yet these models often exhibit discriminatory behavior that negatively impacts protected groups. To address these biases, several debiasing methods have been developed, intervening at different stages of the ML pipeline (d'Alessandro et al., 2017; Mehrabi et al., 2021). Pre-processing debiasing methods, including reweighting, resampling, data augmentation, or even learning fair data representations, modify the training data to remove the underlying discrimination before it is used by the model (Brunet et al., 2019; Kamiran & Calders, 2012). In-processing methods modify the learning algorithms, by, for example, adding constraints to the optimization objective, in order to remove discrimination during the model training process (Zafar et al., 2017c; Zhang et al., 2018). Finally, post-processing methods change the decision thresholds or apply recalibration methods to adjust model outputs to achieve fairness after the model has been trained (Hardt et al., 2016; Bolukbasi et al., 2016). PNKA can be used to analyze any debiasing method that operates at the pre- or in-processing stages of the ML pipeline.

With the increased adoption of debiasing methods, auditing these approaches has received considerable attention. Most fairness measures fall under two main categories (Pessach & Shmueli, 2022): (1) group fairness, which requires parity of some statistical measure across protected groups (Dwork et al., 2012; Hardt et al., 2016). (2) individual fairness, which requires that similar individuals be treated similarly (Joseph et al., 2016; Heidari & Krause, 2018). Both categories are output-based, i.e., they solely look at the predictions/decisions made by the model.

Some work (Gonen & Goldberg, 2019; Caliskan et al., 2017; Bolukbasi et al., 2016; Ethayarajh et al., 2019; Garg et al., 2018) has analyzed representations, mostly in the realm of word embeddings. For instance, Bolukbasi et al. (2016) evaluates gender bias in word embeddings by identifying a gender subspace using explicitly gendered word pairs and analyze the proximity of gender-neutral words to gender-specific terms, uncovering societal stereotypes. Caliskan et al. (2017) uses the WEAT test to systematically measure biases in word embeddings by comparing the association strength between pairs of target words (e.g.,"male" and "female" names) and attribute words (e.g., "career" and "family").

The closest work to ours is the one by Gonen & Goldberg (2019), which investigates word embeddings and shows that debiasing methods for word embeddings mostly hide the bias, instead of removing it. More specifically, using a clustering algorithm, they show words that receive implicit gender from social stereotypes (e.g., receptionist, captain) still tend to group with other implicit-gender words of the same gender, similar as for non-debiased word embeddings. Using PNKA, we reach a similar conclusion: these debiasing models primarily mask, rather than mitigate, the bias. However, as we showed in Section 4, our analysis offers a fresh perspective and instead reveals that these methods are augmenting the gender-related information of the gender-defining words, instead of removing this information from the gender-stereotypical words. None of the previous work provide a general measure that can be applied to representations in different contexts. To the best of our knowledge, our measure is the first one that can be broadly applied to any debiasing method that alters the representations used by models.

### 5.2 Representational Similarity Measures

Several measures have been proposed and used as tools to better understand the internal representations of machine learning (ML) models. Recently, approaches that compare the representational spaces of two models by measuring representational similarity have gained popularity Laakso & Cottrell (2000); Li et al.

(2015); Wang et al. (2018); Raghu et al. (2017); Morcos et al. (2018); Kornblith et al. (2019). At their core, representational similarity measures (RSMs) quantify how a set of points are positioned relative to each other within the representation spaces of two different models. Among the RSMs proposed in the literature, CKA Kornblith et al. (2019) has gained popularity and has now been extensively used to study representations Nguyen et al. (2021); Ramasesh et al. (2020); Raghu et al. (2019; 2021). CKA is based on the idea of first choosing a kernel and then measuring similarity as the alignment between these two kernel matrices. We take inspiration from this insight to propose PNKA.

Most widely used RSMs, however, yield only an aggregate estimate (i.e., a single score) of similarity across an entire set of points. Being limited to aggregate measurements makes these measures unsuitable to study nuances of representational similarity at a more granular, local level. Therefore, approaches of representational similarity for individual points have been proposed, such as for words Hamilton et al. (2016) and nodes in graphs Chen et al. (2021). However, the applicability of these measures is constrained due to their task-specific nature. Work by Shah et al. (2023) proposes a method to estimate the contribution of individual points to learning algorithms, but mainly focuses on understanding what *features* of the inputs are encoded in the representations, and do not evaluate the *similarity* of representations directly. Finally, a pointwise RSM proposed by Moschella et al. (2022) resembles our proposed measure PNKA. However, the goal of Moschella et al. (2022) differs significantly from ours, as they use their measure to enable model stitching, whereas we employ PNKA to audit representations. To the best of our knowledge, this is the first time representational similarity has been employed to study the effects of debiasing methods at a fine-grained level.

## 6 Discussion

We introduce a framework that uses representational similarity to assess how debiasing methods impact individual data representations. Crucial to this framework is PNKA, a pointwise representational similarity measure that quantifies changes in an individual's representation due to fairness interventions. Our findings on datasets like COMPAS and Adult confirm established insights while uncovering new effects, highlighting differences between group- and individual-level fairness. PNKA also enables anticipation of biases and fairness constraints' impacts before model training and deployment, and reveals inconsistencies in how debiasing methods alter targeted groups. Our results suggest that existing fairness metrics are limited, and pointwise representational similarity measures such as PNKA provide a comprehensive tool for auditing fairness interventions at both group and individual levels.

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

# A  Properties of PNKA

## A.1  Overlap of neighbors

In Section 2.1, we argue that for an input example to be similarly represented in two representations, its neighborhood should be similar across both of them. Here, we empirically show that this intuition applies to PNKA, and that if the PNKA score of point $i$ is higher than that of $j$, then $i$'s nearest neighbors in representations $Z$ and $Z'$ overlap more than those of $j$. To show this, we train two models that only differ in their random initialization and compute their representations on the test set (10K instances). We use the penultimate layer (i.e., the layer before logits) for the analysis. For each model, we determine a point's $k$ nearest neighbors by ranking a point's representation distance (via either *cosine similarity* or *L2 distance*) to every other point in that representation. We then compute the fraction of those two sets of $k$ neighbors that intersect.

In the following plots we depicts the relationship between PNKA similarity scores (x-axis) and the fraction of overlapping $k$ nearest neighbors of each point (y-axis), i.e., 1 means all $k$ nearest neighbors are shared between both representations. We report the analysis on CIFAR-10 and CIFAR-100 Krizhevsky et al. (2009), for ResNet-18 He et al. (2016), VGG-16 and Inception-V3, for different $k$ values, up to $k = 20\%$ of the dataset size. All the results are reported over 3 runs. In all cases of Section A.1 we see a clear relationship between high PNKA scores and high overlap of nearest neighbors across representations, indicating that PNKA captures how similar the neighborhoods of the points are. We further show in Figures 8 and 9 that the direct comparison of representations through cosine similarity does not exhibit the same trend, i.e., there is not a positive correlation between cosine similarity scores and the fraction of overlap of the $k$ nearest neighbors.

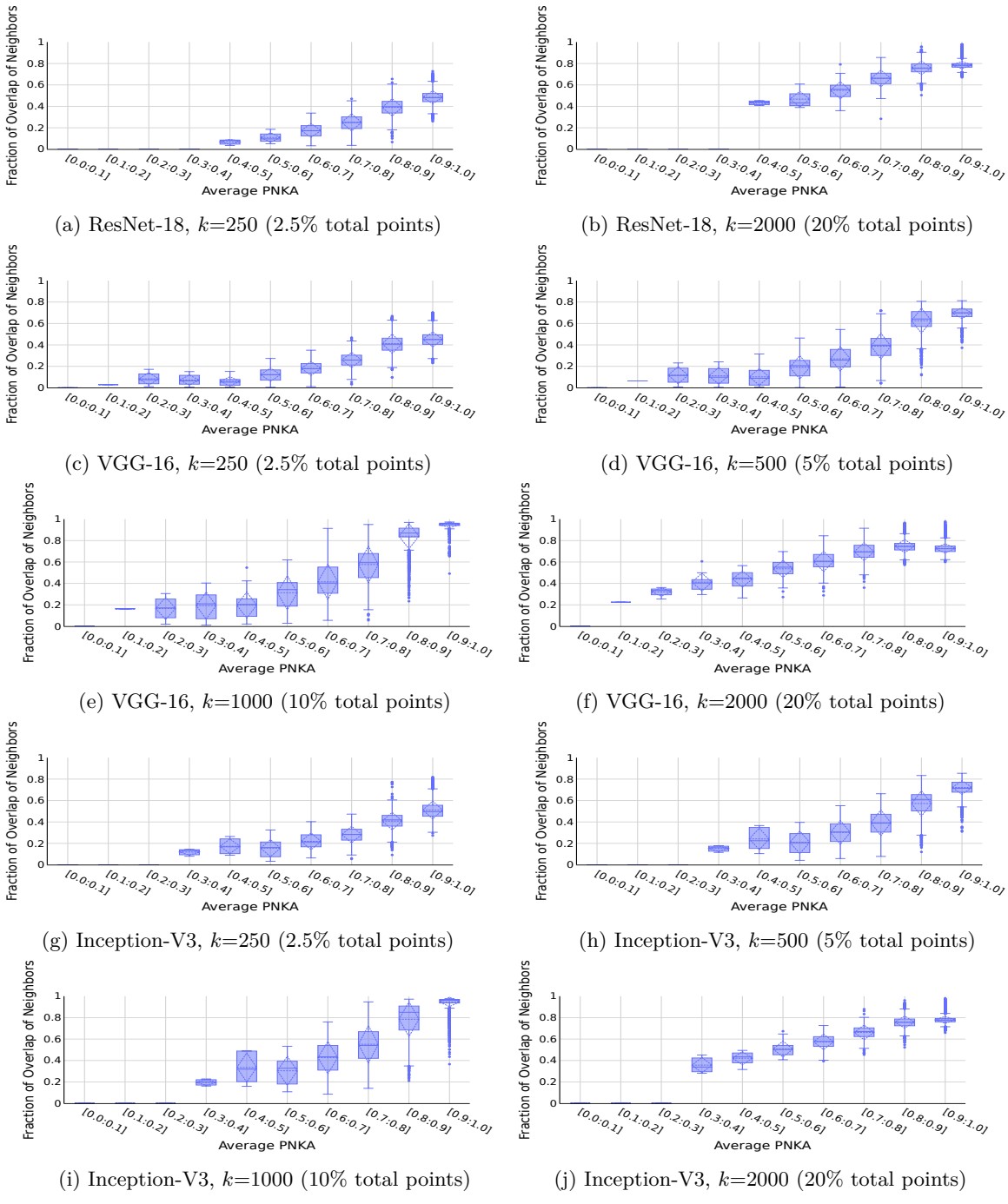

Figure 5: **PNKA** (with linear kernel) captures the overlap of $k$ nearest neighbors between two representations, i.e., the higher PNKA scores, the higher the fraction of overlapping neighbors. Results are an average over 3 runs, each one containing two models trained on **CIFAR-10** Krizhevsky et al. (2009) dataset with the same architecture but different random initialization.

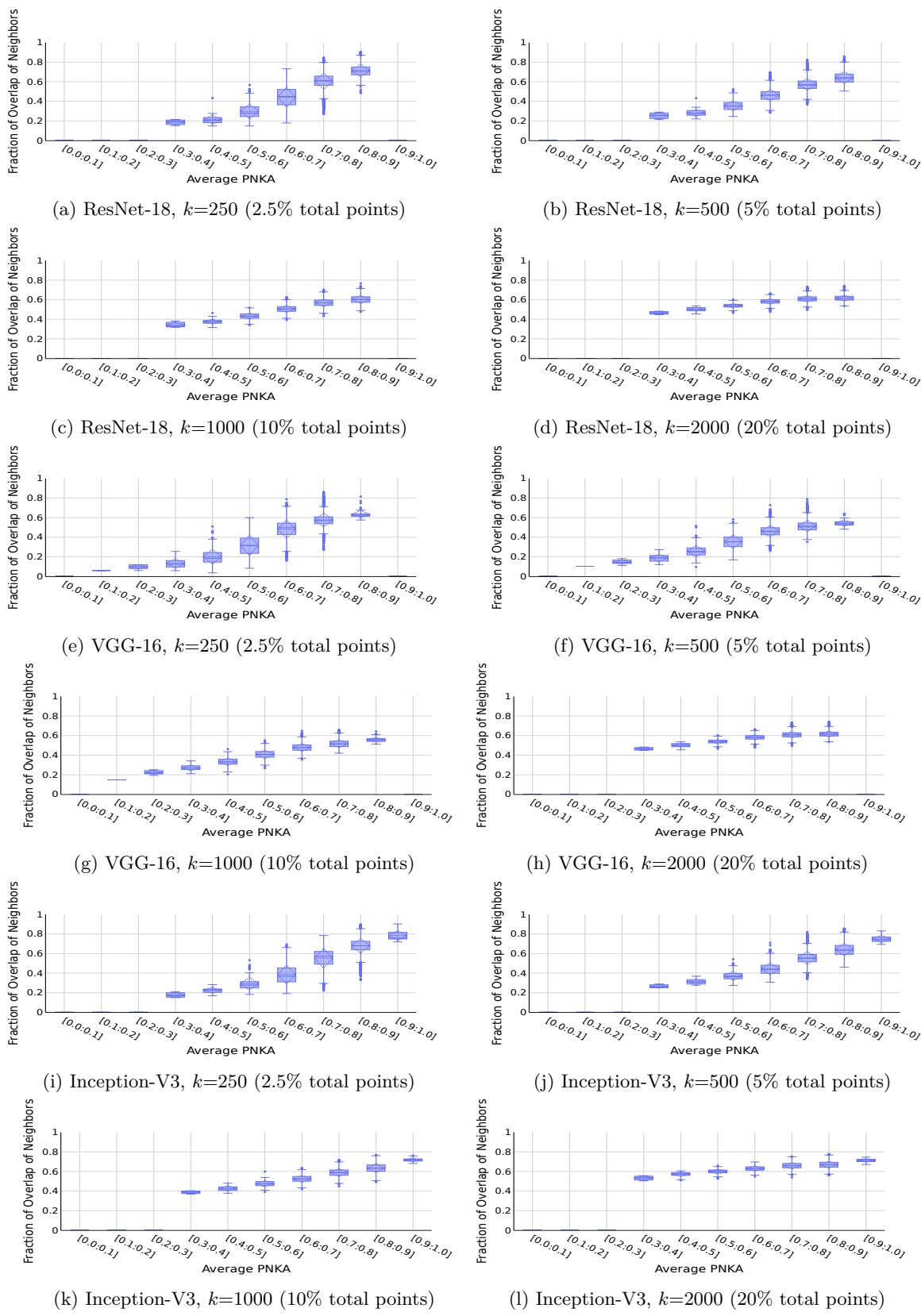

Figure 6: **PNKA** (with linear kernel) captures the overlap of $k$ nearest neighbors between two representations, i.e., the higher PNKA scores, the higher the fraction of overlapping neighbors. Results are an average over 3 runs, each one containing two models trained on **CIFAR-100** Krizhevsky et al. (2009) dataset with the same architecture but different random initialization.

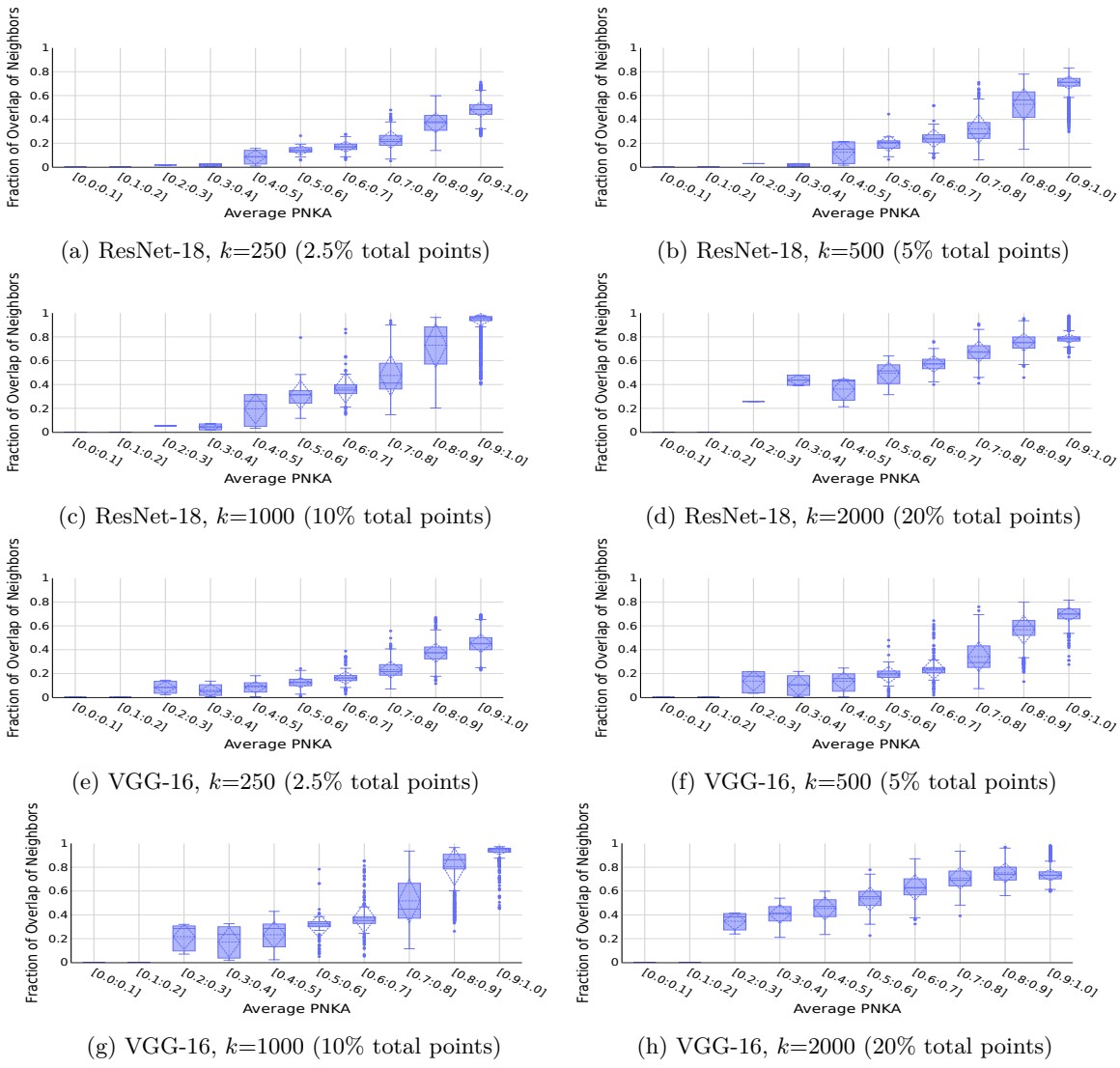

Figure 7: **PNKA** (with RBF kernel) captures the overlap of $k$ nearest neighbors between two representations, i.e., the higher PNKA scores, the higher the fraction of overlapping neighbors. Results are an average over 3 runs, each one containing two models trained on **CIFAR-10** Krizhevsky et al. (2009) dataset with the same architecture but different random initialization.

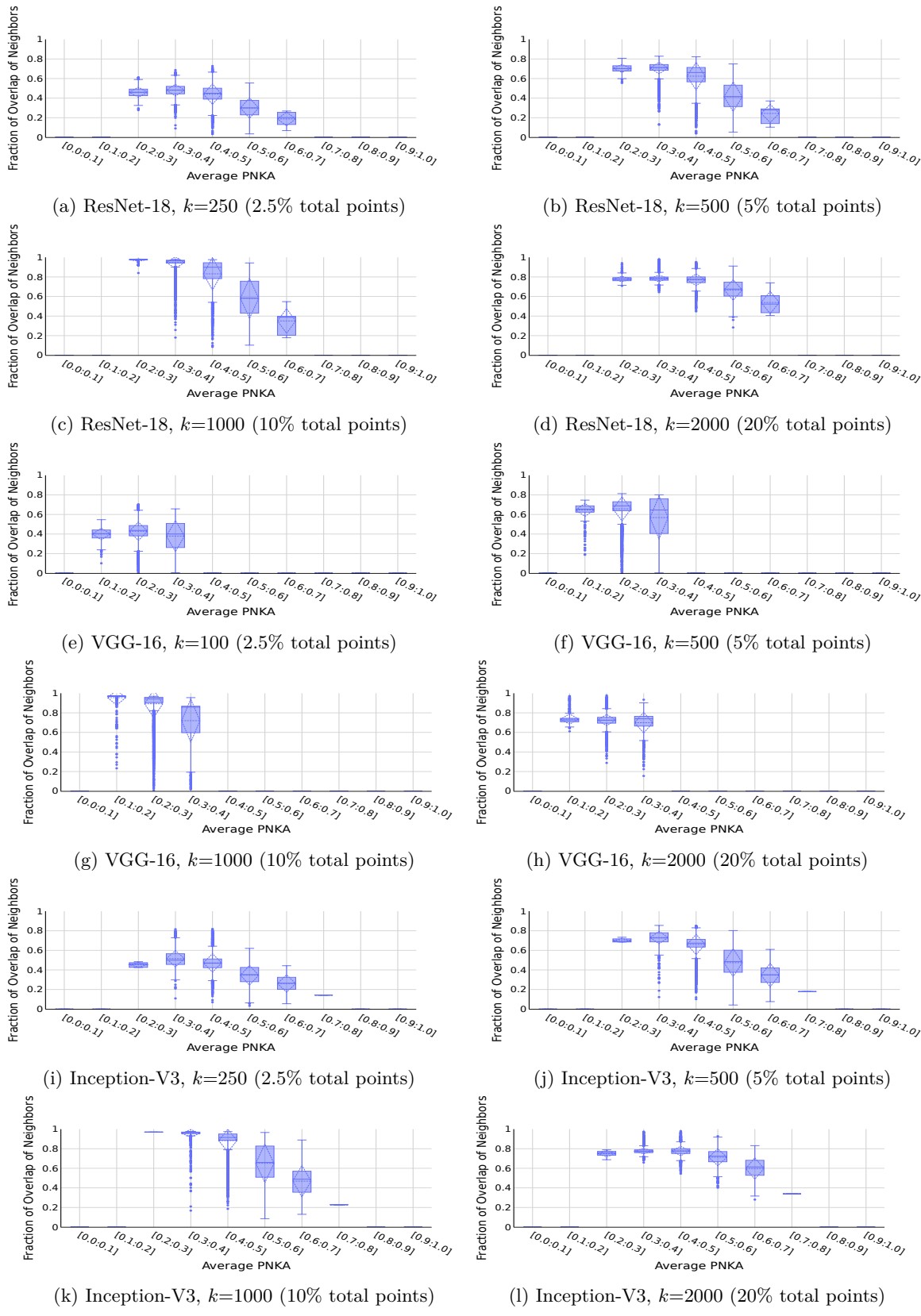

Figure 8: **Cosine similarity** is not able to captures the overlap of $k$ nearest neighbors between two representations i.e., there is not a positive correlation between cosine similarity scores and the fraction of overlap of the $k$ nearest neighbors. Results are an average over 3 runs, each one containing two models trained on **CIFAR-10** Krizhevsky et al. (2009) dataset with the same architecture but different random initialization.

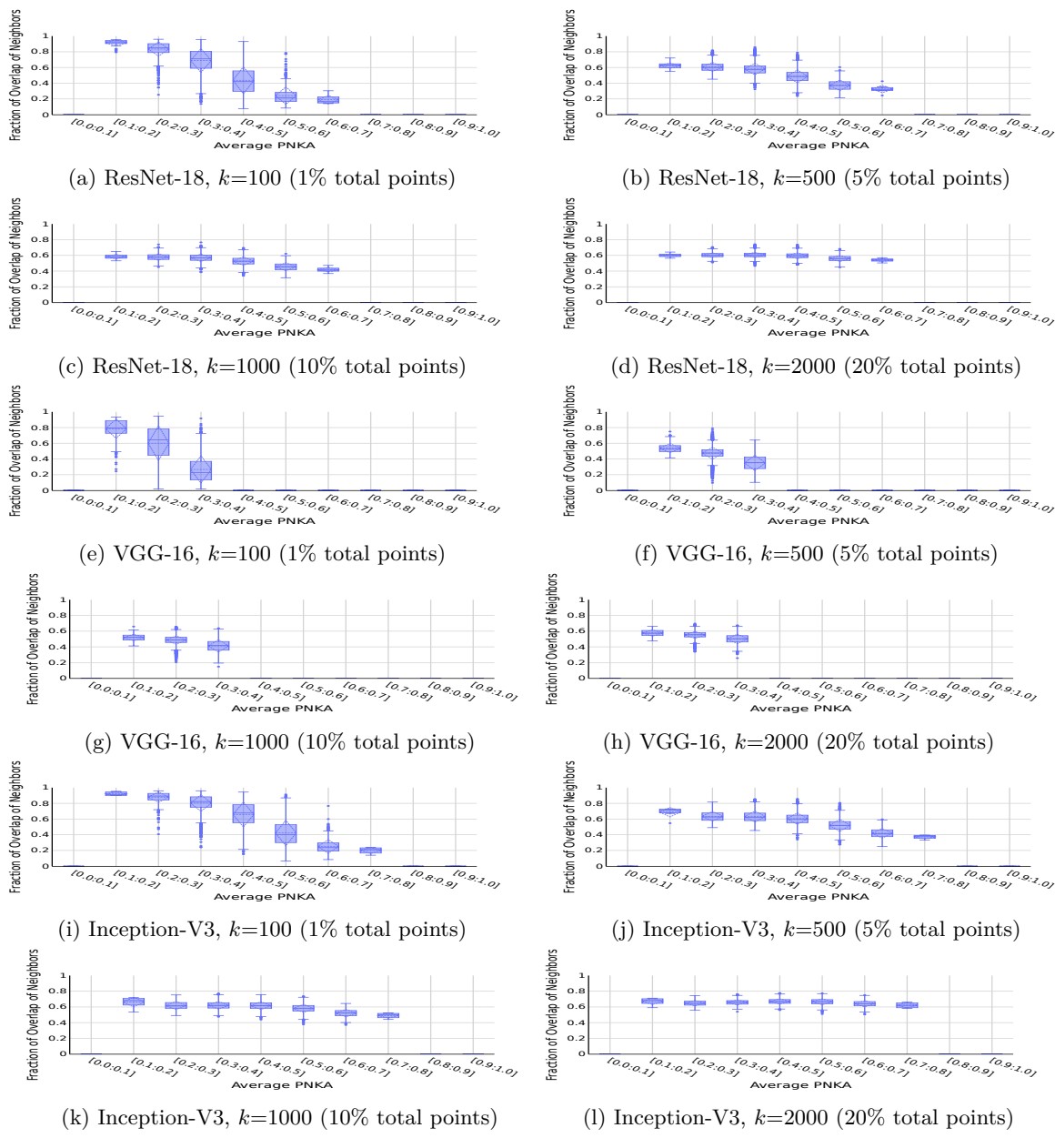

(a) ResNet-18, $k$=100 (1% total points)

(b) ResNet-18, $k$=500 (5% total points)

(c) ResNet-18, $k$=1000 (10% total points)

(d) ResNet-18, $k$=2000 (20% total points)

(e) VGG-16, $k$=100 (1% total points)

(f) VGG-16, $k$=500 (5% total points)

(g) VGG-16, $k$=1000 (10% total points)

(h) VGG-16, $k$=2000 (20% total points)

(i) Inception-V3, $k$=100 (1% total points)

(j) Inception-V3, $k$=500 (5% total points)

(k) Inception-V3, $k$=1000 (10% total points)

(l) Inception-V3, $k$=2000 (20% total points)

Figure 9: **Cosine similarity** is not able to captures the overlap of $k$ nearest neighbors between two representations i.e., there is not a positive correlation between cosine similarity scores and the fraction of overlap of the $k$ nearest neighbors. Results are an average over 3 runs, each one containing two models trained on **CIFAR-100** Krizhevsky et al. (2009) dataset with the same architecture but different random initialization.

## A.2 Relationship of PNKA with aggregate measures of representation similarity

| Dataset | Model | CKA | $\overline{\text{PNKA}}$ |
|---|---|---|---|
| CIFAR-10 | ResNet-18 | 0.925 ($\pm$0.005) | 0.925 ($\pm$0.022) |
| | VGG-16 | 0.895 ($\pm$0.013) | 0.893 ($\pm$0.039) |
| | Inception-v3 | 0.916 ($\pm$0.001) | 0.915 ($\pm$0.023) |
| CIFAR-100 | ResNet-18 | 0.741 ($\pm$0.00) | 0.733 ($\pm$0.033) |
| | VGG-16 | 0.658 ($\pm$0.010) | 0.668 ($\pm$0.049) |
| | Inception-v3 | 0.798 ($\pm$0.009) | 0.792 ($\pm$0.032) |

Table 3: Comparison between CKA Kornblith et al. (2019) and the aggregate version of PNKA ($\overline{\text{PNKA}}$). Results are an average over 3 runs, each one with two models that only differ in their random initialization. We capture the representations of the penultimate layer (i.e., the layer before logits) for the analysis. We show that both measures produce similar overall scores.

## A.3 Proof of invariances

### A.3.1 Invariance to orthogonal transformations

*Proof.* Given an orthogonal matrix $Q$, it suffices to show that

$$
\begin{aligned}
K(ZQ) &= ZQ(ZQ)^\top \\
&= ZQQ^\top Z^\top \\
&= ZQQ^{-1}Z^\top \\
&= ZZ^\top \\
&= K(Z)
\end{aligned}
$$

Here we have used that for an orthogonal matrix $Q$, $Q^\top = Q^{-1}$. By substituting $K(ZQ)$ and $K(Z'R)$ in $\text{PNKA}(ZQ, Z'R, i) = \cos(K(ZQ)_i, K(Z'R)_i)$ with $K(Z)$ and $K(Z')$, respectively, we obtain $\text{PNKA}(ZQ, Z'R, i) = \text{PNKA}(Z, Z', i)$. Thus, PNKA is invariant to orthogonal transformations. $\qquad\square$

### A.3.2 Invariance to isotropic scaling

*Proof.* Note that because of the bilinearity of the dot-product, we have $K(\alpha Z)_i = \left[(\alpha Z)(\alpha Z)^\top\right]_i = \alpha^2 K(Z)_i$. By substituting into PNKA, we get

$$
\begin{aligned}
\text{PNKA}(\alpha Z, \beta Z', i) &= \frac{K(\alpha Z)_i^\top K(\beta Z')_i}{||K(\alpha Z)_i||_2 ||K(\beta Z')_i||_2} \\
&= \frac{\alpha^2 K(Z)_i^\top \beta^2 K(Z')}{||\alpha^2 K(Z)_i||_2 ||\beta^2 K(Z')_i||_2} \\
&= \frac{\alpha^2 K(Z)_i^\top \beta^2 K(Z')}{\alpha^2 ||K(Z)_i||_2 \beta^2 ||K(Z')_i||_2} \\
&= \text{PNKA}(Z, Z', i).
\end{aligned}
$$

Thus, PNKA is invariant to isotropic scaling. $\qquad\square$

# B Investigating Debiased Tabular Data Representations

## B.1 Results with RBF kernel

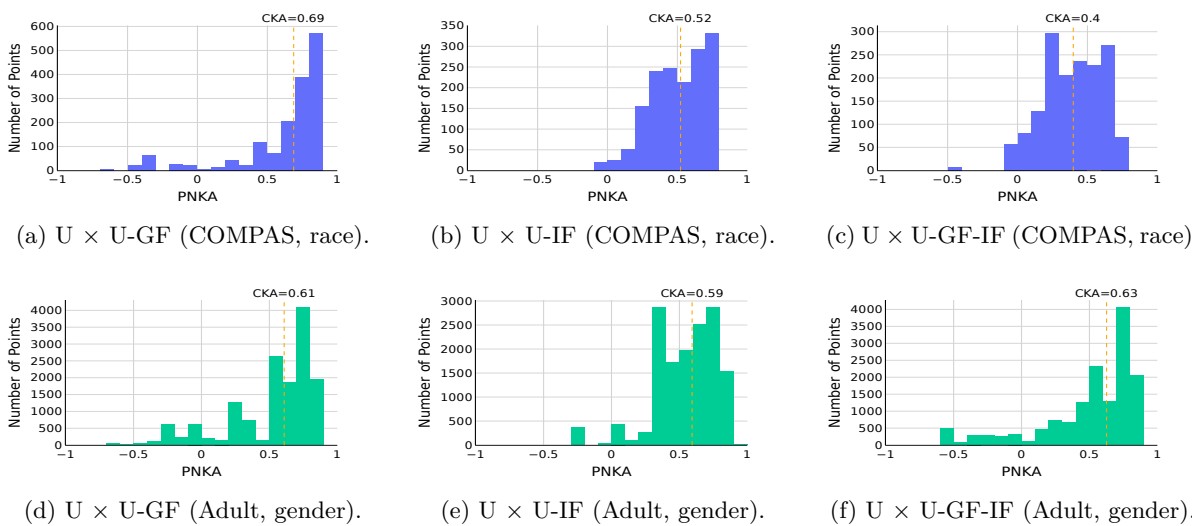

Figure 10: Distribution of PNKA similarity scores with RBF kernel. The first row (blue plots) shows results for the COMPAS dataset, debiased with respect to race, while the second row (green plots) displays results for the Adult dataset, debiased based on gender. The vertical dotted line shows the overall similarity scores provided by CKA Kornblith et al. (2019). We compare the representations obtained from the baseline (U, only utility) with each of the three other debiased options: (1) utility, group and individual fairness (U-GF-IF), utility and group fairness (U-GF), and utility and individual fairness (U-IF).

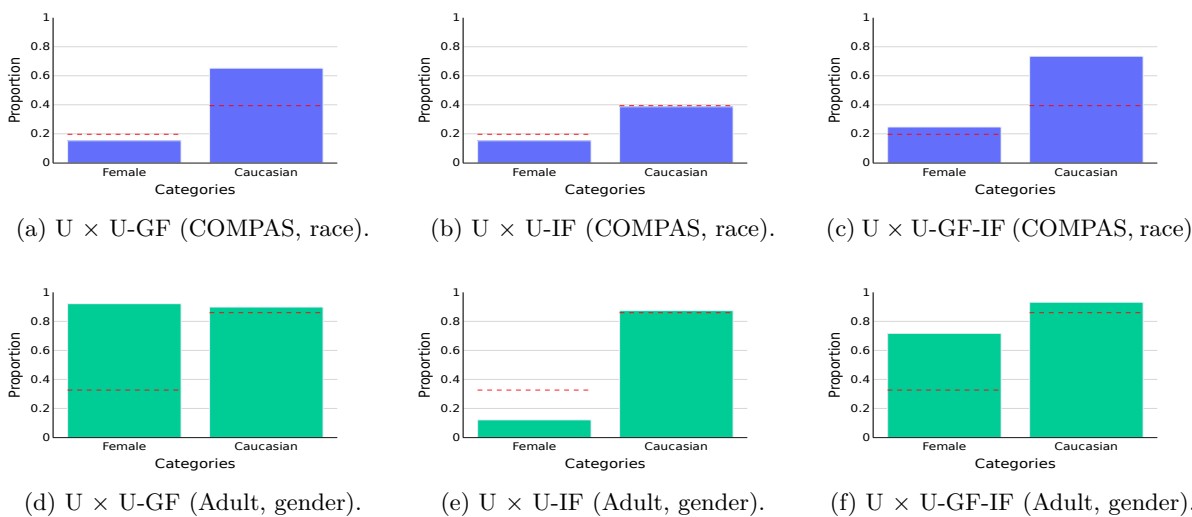

Figure 11: Distribution of the binary attributes for the 10% most affected individuals (i.e., lowest PNKA score) using the RBF kernel. The first row (blue plots) shows results for the COMPAS dataset, debiased with respect to race, while the second row (green plots) displays results for the Adult dataset, debiased based on gender. We compare the data representations of the baseline (U, only utility) with each of the three other debiased options: (1) utility, group and individual fairness (U-GF-IF), utility and group fairness (U-GF), and utility and individual fairness (U-IF).

## B.2 Distribution of attributes for the data points whose representation changed the most

In this section, we present graphical illustrations of the distribution of all the attributes within the COMPAS and Adult datasets using a series of concentric rings. Each ring functions as a pie chart, depicting the distribution of a specific attribute. For instance, the innermost two rings represent the gender (male/female) and racial (Caucasian/non-Caucasian) distributions in both datasets. The subsequent rings are tailored to each dataset. For the COMPAS dataset, the third, fourth, and fifth outer rings display the distributions of age categories (under 25, 25-45, over 45 years old), prior counts (0, 1-3, more than 3), and type of charge (felony or misdemeanor), respectively. For the Adult dataset, the third and fourth outer rings showcase the age categories (grouped by decades) and education levels. Additionally, the first plot of Figure 12 and Figure 13, located in the top left, illustrates the overall distribution of these attributes across the entire dataset. The subsequent plots provide a detailed view of the attribute distributions for the bottom 10% of data points, which exhibited the most significant changes in their representation.

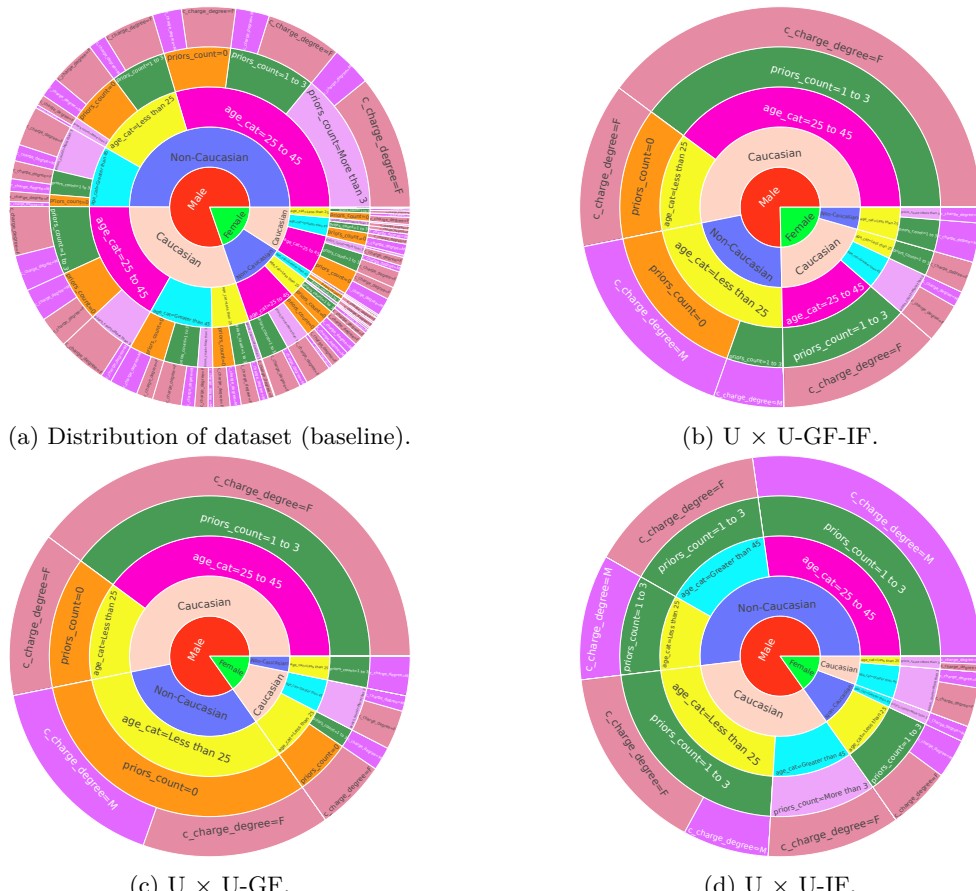

(a) Distribution of dataset (baseline).

(b) U × U-GF-IF.

(c) U × U-GF.

(d) U × U-IF.

Figure 12: Distribution of all the attributes of the 10% of instances with the lowest PNKA scores for COMPAS dataset with the protected attribute as race.

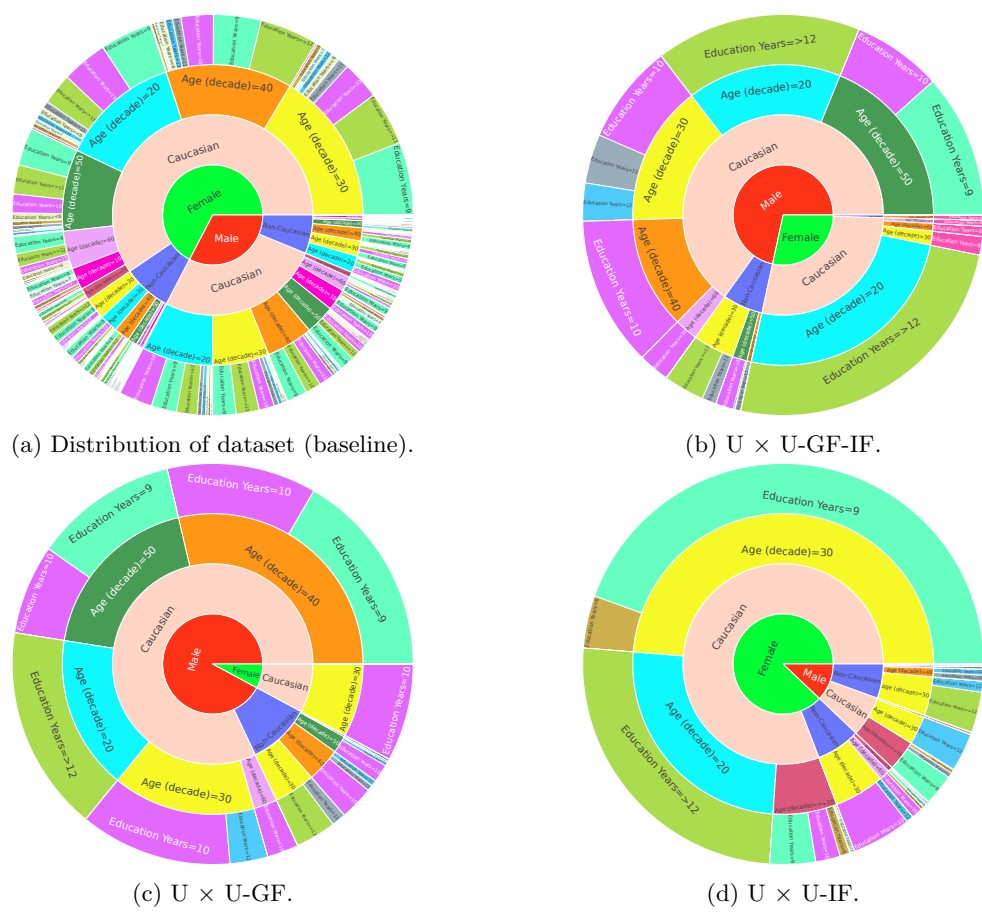

(a) Distribution of dataset (baseline).

(b) U × U-GF-IF.

(c) U × U-GF.

(d) U × U-IF.

Figure 13: Distribution of all the attributes of the 10% of instances with the lowest PNKA scores for Adult dataset with the protected attribute as gender.

## C Investigating Debiased Word Embeddings

### C.1 PNKA results with RBF kernel

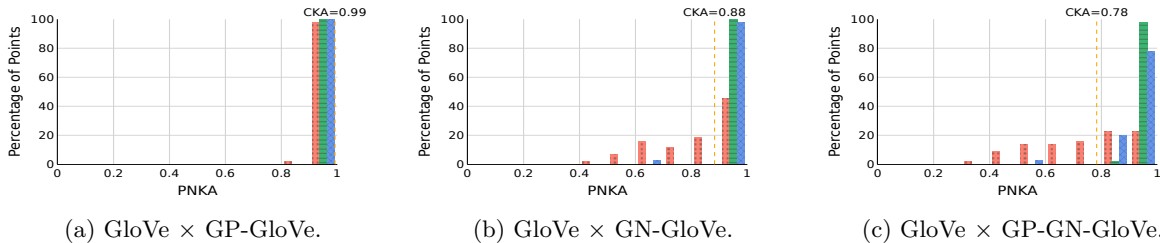

(a) GloVe × GP-GloVe.    (b) GloVe × GN-GloVe.    (c) GloVe × GP-GN-GloVe.

Figure 14: Distribution of PNKA scores (with RBF kernel) per group of words for SemBias dataset Zhao et al. (2018). We compare the baseline (GloVe) model and its debiased versions. Words with the lowest similarity scores are the ones that change the most from the baseline to its debiased version. Across all debiased embeddings, the words whose embeddings change the most are the gender-definition words.

### C.2 Evaluation of Debiasing Methods on SemBias dataset

For each of the four word pairs $(a, b)$ in a SemBias instance, GP- and GN-Glove measure its cosine similarity with the canonical gender vector, i.e., $cos(\overrightarrow{a} - \overrightarrow{b}, \overrightarrow{he} - \overrightarrow{she})$. The word pair with the highest cosine similarity is selected as the "predicted" answer. If the word embeddings are correctly debiased, then the cosine similarity of the $\overrightarrow{he}$ - $\overrightarrow{she}$ vector with the gender-definition words should be high, and the similarity with the gender-stereotype words should be low, i.e., the frequency of predictions for these categories should be high and low, respectively. Table 4 depicts the results for the GN- and GP-Glove (Zhao et al., 2018; Kaneko & Bollegala, 2019) methods.

| Embeddings | SemBias | | |
|---|---|---|---|
| | Definition ↑ | Stereotype ↓ | Neutral ↓ |
| GloVe | 80.2 | 10.9 | 8.9 |
| GN-GloVe | 97.7 | 1.4 | 0.9 |
| GP-GloVe | 84.3 | 8.0 | 7.7 |
| GP-GN-GloVe | 98.4 | 1.1 | 0.5 |

Table 4: Frequency of predictions for gender relational analogies (Kaneko & Bollegala, 2019). Each column shows the frequency with which the respective word-pair category (gender-definitional, gender-stereotype, gender-neutral) is predicted as having the highest cosine similarity with the canonical gender vector $\overrightarrow{he}$ - $\overrightarrow{she}$. The more often gender-definition words are predicted as being most gender-aligned, as opposed to gender-stereotype words, the less biased an embedding approach can be considered.

# D   Investigating Debiased Contextual Embeddings

Transformer-based language models are currently the leading approach for many NLP tasks. In contrast to the previously discussed word embeddings, these models use contextualized embeddings where the representation of a word or token also depends on the preceding and possibly succeeding tokens. However, just like their non-contextualized predecessors, these models have also been found to embed several unfair biases (Zhao et al., 2019; Tan & Celis, 2019; Kurita et al., 2019). The predominant approach to using transformer-based (large) language models is to adapt pre-trained models to downstream tasks, either with or without fine-tuning. Any biases that the pre-trained base-models exhibit might thus be inherited by the downstream tasks. Since a small number of base-models are adapted for many different downstream tasks, it is infeasible to audit models on each of those tasks separately. Rather, analyzing biases in the representations that base models use provides a scalable way to study their fairness.

Given the importance of addressing these biases for transformer-based language models, researchers have started proposing methods aimed to debias the contextualized embeddings of these models. One method by Kaneko & Bollegala (2021) proposes to remove a potential gender bias by fine-tuning a contextual baseline model to "preserve semantic information with respect to sentences with sensitive attributes (e.g., 'she', 'he'), while removing any discriminatory biases with respect to sentences with stereotypical words (e.g., 'poetry', 'math')". In other words, the goal of this method is to remove gender bias from the representations of sentences containing potentially stereotypical words, such as ones related to poetry (with a presumed female bias) and ones related to math (with a presumed male bias).

We leverage PNKA to investigate whether this debiasing method changes the representations of sentences in the intended manner. For our analysis, we use the Albert ('albert-base-v2') model as baseline, and evaluate it and its debiased versions on the same dataset used to study the gender bias in the original work of Kaneko & Bollegala (2021), specifically the SEAT-7 and SEAT-8 datasets (May et al., 2019). These datasets are composed of simple sentence templates such as "This is a [BLANK]", and create gender defining and gender stereotypical sentences by substituting "[BLANK]" with gender defining (e.g., 'she', 'he') and target stereotypical words (e.g., 'poetry', 'math'), respectively. The specific words were chosen based on the WEAT measure (Caliskan et al., 2017), which measures the associations between concepts like math/art and science/art with male/female attributes in SEAT-7 and SEAT-8, respectively. A more detailed explanation on WEAT and SEAT is provided in Appendix D.1 and D.2.

Figure 15 shows the distribution of PNKA similarity scores for the two categories of sentences in SEAT-7 (Figure 15a) and SEAT-8 (Figure 15b), respectively. In both cases, we observe an overall high PNKA score, with more than 80% of points obtaining PNKA scores higher or equal to 0.8, and CKA of 0.88. This suggests that most of the instances of both gender defining as well as gender stereotypical sentence categories have not drastically changed from the baseline to the debiased model. More surprisingly, there is no clear distinction between the two groups of sentences, i.e., both gender defining (e.g., "This is a woman." or "This is a man.") and gender stereotypical (e.g., "This is poetry." or "This is math.") sentences change in a similar proportion.

To investigate whether the high representation similarity indicates a lack of change in the gender properties of the sentence, we follow the procedure used for the non-contextual embeddings described in Section 4. Again, we project the sentence representations onto the gender vector $\overrightarrow{he}$ - $\overrightarrow{she}$ [6] and measure the change in projection magnitude from the baseline to the debiased version. We follow Kaneko & Bollegala (2021) and obtain representations of the [CLS] token at the last layer. Figure 16 displays the relationship between PNKA scores and the percentage difference $\omega_i^{(baseline)}$ for the baseline ('albert-base-v2') as well as $\omega_i^{(debiased)}$ for debiased model. In accordance with the high PNKA scores, we observe that all the contextual embeddings exhibit a minimal shift along the gender direction. Once again, PNKA is a reliable indicator that the debiasing technique is not effective in the intended way, and that the bias of gender stereotypical sentences is not reduced as expected. Our insights complement previous work, which also identified that using SEAT alone is insufficient to evaluate debiasing methods (Meade et al., 2021; Silva et al., 2021; May et al., 2019). Future work could explore the effects of this method further to understand why it does not significantly alter the representations in the intended way.

---

[6]In Appendix D.3 we also explore using the average contextual gender vector and observe a similar pattern.

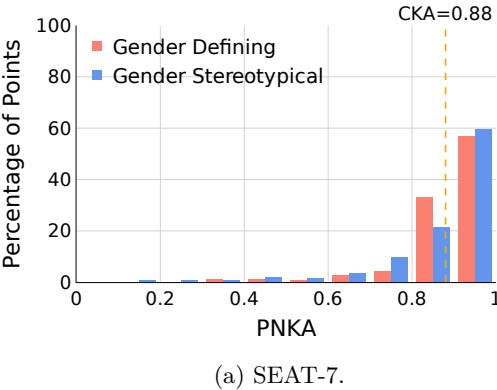 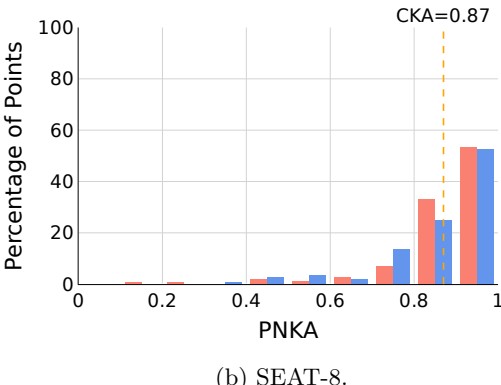

(a) SEAT-7.          (b) SEAT-8.

Figure 15: Distribution of PNKA scores (linear kernel) per group of sentences in SEAT-7 and SEAT-8 dataset May et al. (2019). We compare the baseline ('albert-base-v2') model and its debiased version (Kaneko & Bollegala, 2021). Sentences with the lowest similarity scores are the ones that change the most from the baseline to the debiased version. Across all the datasets, we observe that most of the sentences maintain high PNKA scores, which indicates that they have not substantially changed their representations. Moreover, there is no clear difference between the groups of gender defining and gender stereotypical sentences in how they change their representations.

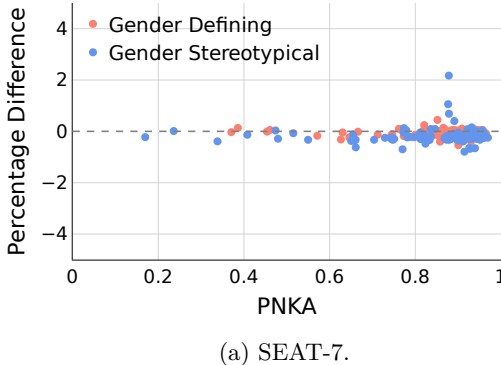 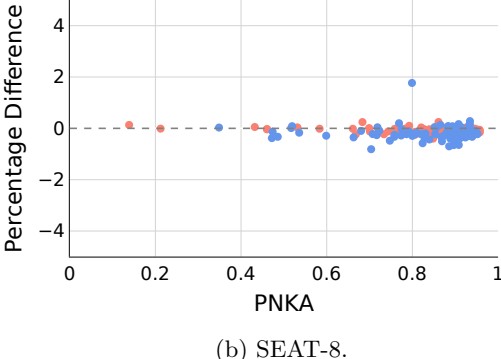

(a) SEAT-7.          (b) SEAT-8.

Figure 16: Relationship between PNKA scores (x-axis) and percentage difference (y-axis) in magnitude of the projection on the gender direction $\overrightarrow{he}$ - $\overrightarrow{she}$. A positive or negative percentage difference value indicates a shift in magnitude along the gender direction. Overall, the contextual embeddings exhibit a low shift along the gender direction, with no clear distinction between groups of sentences.

## D.1 Word Embedding Association Test (WEAT)

The description below is provided by Li et al. (2023). Word Embedding Association Test (WEAT) (Caliskan et al., 2017) measures the association between two sets of attributes words (e.g., male and female) and two sets of targets words (e.g., family and career). Formally, the sets of attribute words are indicated by $A$ and $B$, and the sets of target words are denoted by $X$ and $Y$. Then, the WEAT test is as follows:

$$s(A, B, X, Y) = \sum_{x \in X} s(x, A, B) - \sum_{y \in Y} s(x, A, B), \qquad (2)$$

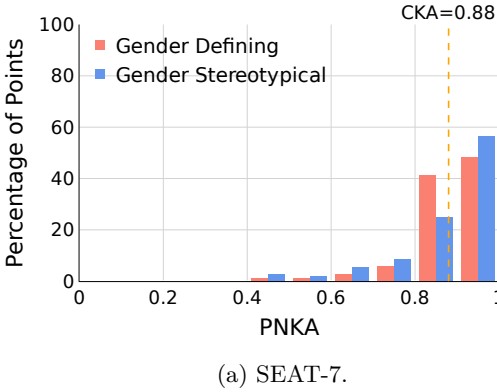

(a) SEAT-7.

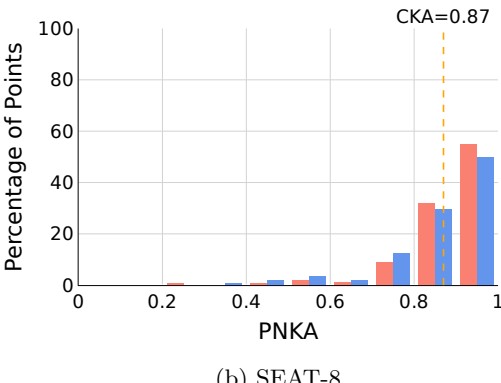

(b) SEAT-8.

Figure 17: Distribution of PNKA scores (RBF kernel) per group of sentences in SEAT-7 and SEAT-8 dataset May et al. (2019). We compare the baseline ('albert-base-v2') model and its debiased version (Kaneko & Bollegala, 2021). Sentences with the lowest similarity scores are the ones that change the most from the baseline to the debiased version. Across all the datasets, we observe that most of the sentences maintain high PNKA scores, which indicates that they have not substantially changed their representations. Moreover, there is no clear difference between the groups of gender defining and gender stereotypical sentences in how they change their representations.

where $s(w, A, B)$ represents the difference between the average of the cosine similarity of word w with all words in $A$ and the average of the cosine similarity of word w to all words in $B$, and it is defined as follows:

$$s(w, X, Y) = \frac{1}{|A|} \sum_{a \in A} cosine(w, a) - \frac{1}{|B|} \sum_{b \in B} cosine(w, b), \tag{3}$$

where $w \in X$ or $Y$, and $cosine(\cdot, \cdot)$ represents the cosine similarity. The normalized effect size is as follows:

$$d = \frac{\mu(\{s(x, A, B)_{x \in X}\}) - \mu(\{s(y, A, B)_{y \in Y}\})}{\sigma(\{s(t, X, Y)_{t \in A \cup B}\})}, \tag{4}$$

where $\mu(\cdot)$ is the mean function and $\sigma(\cdot)$ is the standard deviation.

## D.2 Sentence Embedding Association Test (SEAT)

Sentence Embedding Association Test (SEAT) (May et al., 2019) adapts WEAT to contextual embeddings, which uses simple sentence templates such as "This is a [BLANK]" to substitute attribute words and target words to obtain context-independent embeddings. Then the SEAT test statistic between the two sets of embeddings (represented by the '[CLS]' of the last layer) is calculated similarly to Equation 4.

### D.3 Results of Projection with Average Contextual Gender Direction

To investigate whether the high representation similarity indicates a lack of change in the gender properties of the sentence, we follow the procedure used for the non-contextual embeddings described in Sections 4. However, in this case, instead of just using the gender directions of the words *he* and *she*, we use a gender direction from an aggregated embeddings of male and female sentences. In other words, we project the sentence representations of the SEAT dataset onto the average contextual gender vector originally used by Kaneko & Bollegala (2021) to debias the model. This average contextual gender vector is obtained as follows. First, we obtain an average representation for the sentences $S(w)$ where the gendered-word $w$ appears. This gendered-word $w$ belongs to an attribute $A \in \mathcal{A}$, where $\mathcal{A} = \{A_f, A_m\}$, and $A$ denotes a set of words associated with the corresponding gender, where $f$ stands for female and $m$ stands for male.

$$e_A(w) = \frac{1}{|S(w)|} \sum_{s \in S(w)} e(s). \tag{5}$$

Next, we take the final average contextual gender vector for attribute $A$, which can be $A_f$ for female and $A_m$ for male, as follows:

$$\overline{e}_A = \frac{1}{|A|} \sum_{w \in A} e_A(w). \tag{6}$$

Thus, the projection will be computed using the average contextual gender direction $\overline{e}_{A_m} - \overline{e}_{A_f}$.

Figure 18 displays the relationship between PNKA scores and the percentage difference for the baseline ('albert-base-v2') as well as for debiased model according to the average contextual gendered direction. As before, and in accordance with the high PNKA scores, we observe that most of the contextual embeddings exhibit a minimal shift along the gender direction.

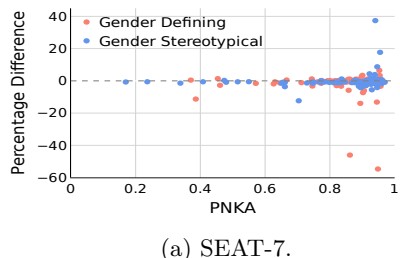

(a) SEAT-7.

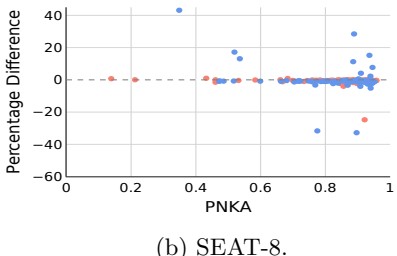

(b) SEAT-8.

Figure 18: Relationship between PNKA scores (x-axis) and percentage difference (y-axis) in magnitude of the projection on the average gendered direction. A positive or negative percentage difference value indicates a shift in magnitude along the gender direction. Overall, the contextual embeddings exhibit a low shift along the gender direction, with no clear distinction between groups of sentences.

