# OpenReview forum: "Investigating the Effects of Fairness Interventions Using Pointwise Representational Similarity"
_TMLR — Accepted by TMLR_

### Review · Reviewer_tRz9 · 2024-11-20

**Summary Of Contributions:**

This submission addresses the limitations of existing fairness evaluation measures for machine learning (ML) models, particularly their lack of granularity and task generalizability. The authors propose Pointwise Normalized Kernel Alignment (PNKA), a novel metric that analyzes how fairness interventions impact the intermediate representations of individual data points, offering insights into both group-level and individual-level fairness. Unlike conventional evaluation methods, PNKA allows for a fine-grained assessment by measuring representational similarity at the individual level. The study reveals that group fairness interventions often affect only a small subset of individuals while maintaining representational similarity for most of the population, whereas individual fairness constraints lead to broader, more uniform changes across data points. Additionally, PNKA is shown to predict downstream task behavior accurately and highlights unintended shortcomings of existing debiasing methods in language embeddings, such as their failure to effectively remove biases from stereotypical words and sentences. These findings emphasize the inadequacy of current fairness evaluation measures and underscore the utility of pointwise representational similarity metrics, such as PNKA, for deeper fairness audits across a range of ML applications.

**Audience:**

Yes

**Broader Impact Concerns:**

I do not have any concerns regarding ethical implications.

**Claims And Evidence:**

Yes

**Requested Changes:**

I have no changes to request, since I found this submission overall well-written.

**Strengths And Weaknesses:**

The paper is overall well-written and sound. The authors provide a clear rationale for introducing PNKA and effectively connect it to existing fairness challenges, helping readers understand the relevance and significance of the proposed method. The related work is discussed in detail, and the experiments are clearly explained. Overall, this submission is accessible to a broad audience. Furthermore, the problem studied in this submission is relevant, and of high interest for the ML community. The proposed metric, which measures similarity on the representation space, is a novel approach, and the experiment on language embeddings are particularly interesting. My only concern is the lack of a technical contribution in this work, since the main contribution is just Eq. (1).

---

> ### Author Response · Authors · 2024-11-29
>
> Thank you for reviewing our paper and for your thoughtful feedback.
>
> While Eq. (1) introduces the PNKA measure as a technical innovation, our main contribution extends beyond this. It *also* lies in demonstrating how PNKA can be effectively applied to *analyze the impact of fairness interventions on representations at the individual level across multiple domains*. This application highlights the broader utility of PNKA as a critical tool for advancing fairness research.
>
> We hope this addresses your concern, and we are happy to provide further clarification if needed.

---

> > ### Comment · Reviewer_tRz9 · 2024-11-29
> >
> > Thank you for your reply. Let me then rephrase my question: What technical challenges did you encounter, when you analyzed the impact of fairness interventions on representations at the individual level across multiple domains?

---

> > > ### Author Response · Authors · 2024-12-08
> > >
> > > Thank you for your question. The main technical challenges we encountered when analyzing the impact of fairness interventions on representations at the individual level across multiple domains are as follows:
> > >
> > > 1. **Designing PNKA (Section 2)**: Designing PNKA involved balancing simplicity and interpretability while also holding key mathematical properties such as invariance to isotropic scaling and orthogonal transformations. The challenge was to ensure that the measure captured meaningful differences in individual representations across diverse datasets and scenarios. This required theoretical (section 2.3) as well as experimental (sections 3 and 4) validations to avoid misleading results due to transformations and careful testing to ensure generalizability. Balancing these considerations while avoiding the pitfalls of existing global similarity measures was very important to PNKA’s design.
> > >
> > > 2. **Tabular Data (Section 3)**: The primary goal here was to demonstrate how PNKA can audit the effects of debiasing approaches on widely studied cases. This served as a sanity check to validate that the insights provided by PNKA were both reliable and consistent with findings from prior outcome-based studies. Most importantly, we wanted to show that PNKA could be applied efficiently and generically, without requiring a specific downstream application. The main technical challenge, therefore, was in designing the right experimental setup to ensure our formulation of PNKA aligned with these goals and reliably captured meaningful insights.
> > >
> > > 3. **Language Embeddings (Section 4)**: In contrast, this section aimed to show that PNKA could provide novel insights into domains with more specialized and non-generalizable evaluation metrics. Specifically, we highlighted how PNKA revealed an alternative explanation for what these debiasing methods might be doing -- amplifying gender information in gender-defining words, rather than effectively removing bias in gender-stereotypical words, as previously claimed. The key technical challenge was formulating a hypothesis to explain the differences between our observations and prior claims,  and subsequently integrating these findings into the broader understanding, while uncovering an important aspect that previous studies may have overlooked. This required PNKA to reveal subtle but critical insights that existing tools could not provide, emphasizing its novelty and domain-specific adaptability.
> > >
> > > We hope this clarifies the challenges we faced, and we would be happy to provide additional details or address further questions if needed.

---

### Review · Reviewer_twWV · 2025-01-14

**Summary Of Contributions:**

The paper proposes a tool for evaluating the similarity of two representations based on a point. That is how similarly both representations represent a point. The method is inspired by previous work that computes an aggregate similarity measure of two representations. The paper empirically shows, across some datasets, that their proposed method satisfies some nice properties such as having extent of intersection of k-nearest neighbors correspond positively with similarity in representation, and invariance to some transformations.

The paper shows applications of using individual, point-based similarity measures that can’t
be solved using aggregate similarity measures. The applications shown in the paper are: identifying subgroups that change through debiasing techniques on representations, predicting downstream consequences of representations debiasing, and auditing debiasing techniques for word embeddings.

**Audience:**

Yes

**Claims And Evidence:**

Yes

**Requested Changes:**

To strengthen the paper:

Provide what kernel was used and analyze the effects of using other kernels.

Comparison with other individual similarity metrics that are induced by average similarity measure.

**Strengths And Weaknesses:**

Strengths: demonstrating the usefulness of individual representation similarity measures for more fine-grained analysis.

Weaknesses: The method itself does not seem different from previous methods. In fact, it seems that every previous aggregate representation method is averaging over an implicit individual similarity measure.

The kernel used to compute the similarity measure for the empirical results is unclear. It is also unclear what happens if a different kernel is chosen. And if the choice of the kernel changes outcomes significantly, it is unclear how to appropriately choose the kernel.

---

> ### Author Response · Authors · 2025-01-24
>
> We thank the reviewer for their feedback.
>
> **Clarification on kernel usage:** We will explicitly clarify in the revised version of the paper that we used the linear kernel throughout all our analyses. We opted for the linear kernel due to its simplicity and the fact that the linear kernel is widely used in the literature. However, we have also tested our method with the RBF kernel and found that the conclusions remain consistent. We will include the experiments with the RBF kernel and clarify these points in a revised version of the paper, which will be uploaded once all reviews are in.
>
> **Comparison with other individual similarity metrics:** We appreciate the reviewer raising this point. However, we want to emphasize that our main contribution lies in demonstrating the usefulness of applying individual representation similarity measure to fairness analysis, a direction unexplored in prior work. While our method was inspired by previous aggregate similarity measures, it extends the concept to compute individual similarity scores with essential properties such as invariance to isotropic scaling and orthogonal transformations. Our focus is on the method’s applicability, specifically in identifying individuals most affected by debiasing techniques and predicting their downstream consequences, serving as a framework for auditing these methods.
> We've already demonstrated the method's robustness through: (1) Correlation with nearest neighbor overlap; (2) Alignment with aggregate representation similarity measure; (3) Consistency across different kernels (linear and RBF).
> The empirical results show that the individual similarity approach provides more fine-grained insights than aggregate measures, which is the main focus of our work. We are happy to include additional comparative analyses if there are specific aspects you would like us to address.
>
> We hope this addresses your questions and are happy to provide further clarification if needed.

---

### Review · Reviewer_GnRZ · 2025-01-29

**Summary Of Contributions:**

This paper proposes to study the effect of debiasing methods on the intermediate representations of individual training data points.

1) They use a measure of representation similarity to assess the impact of fairness procedures on individual data points before and after debiasing. The similarity measure used is based on the relative position of the data points compared to the other training points within the same representation space. Specifically, an individual data point's representations before and after debiasing of the model are deemed similar if the data point maintain the same position relative to the other training points before and after debiasing.

2) The author use this metric to study the overall individual effect of a group fairness and individual fairness procedure proposed by Zemel et al 2013 and debiasing of word embeddings as proposed by Kaneko and Bollegala 2019. By examining the distribution of similarity scores before and after debiasing as well as the distribution of data points with the lowest scores (indicating the most impacted by the debiasing procedure), they provide several insights into how the debiasing methods studied affect the representation of individual data points.

3) Finally, the authors propose using the results of the similarity measure to understand performance on subsequent tasks using the representations.

**Audience:**

Yes

**Claims And Evidence:**

Yes

**Requested Changes:**

Request:
- Expand the discussion on the validity of using position as a proxy for similarity

Minor changes:
- In section 3.1, the description of figures 1c and 1e is confusing. Is it the other way around?: the pattern for the combination of group fairness and individual fairness for COMPAS resembles that of individual fairness alone and for Adult, it mirrors the distribution when enforcing group fairness only
- Add the legend for the red dotted line on the caption

**Strengths And Weaknesses:**

The main strength of the paper resides in its clear and well plan out investigation. Indeed, the paper is clearly written and well motivated. The investigation is well thought out and the experiments are well executed. I think the overall question of how individual representations of data points are impacted by debiasing methods is somewhat interesting with the application to understanding how models trained on these representations will perform.

In terms of weaknesses, I have 3 main concerns:
1) While the authors have shown how looking at the changes in representations can inform downstream performance, it is unclear what novel insight this provides. For example, in the first example of group and individual fairness for tabular data, the insights around how group and individual fairness technique impact individual representations are not surprising. Indeed, it is expected that methods to satisfy individual fairness would lead to most changes in the representation of most data points compared to group fairness technique which would mostly impact the subset of points that lead to group disparities. Similarly for the insights on using the similarity score to understand performance, the results are not surprised. It can be expected that the debiasing method will increase utility for the minority group.

2) Proof for using relative position as the similarity metric: while the authors provide an intuitive understanding of why they use the relative position of the data point as the similarity metric between the representation before and after debiasing, it is unclear why this is the right choice of metric. The second paragraph of section 2.1 should be expanded to clearly address why this is a correct measure of similarity.

3) Limited methods: the experiments mainly look at one method for each of the case studies. For each case, it is unclear how the insights generalize to other debiasing methods. For example, the experiments on group and individual fairness only look at Zemel et al 2013 which is only an example of method for learning fair representations. Looking at a broader set of fair representation learning methods and individual fairness methods can help assess the validity the insights provided.

---

> ### Author Response · Authors · 2025-02-05
>
> We thank the reviewer for the thoughtful feedback. We address each of the main concerns below and outline how we plan to revise our paper accordingly. Minor changes will be incorporated into the revised manuscript.
>
> 1. **Novelty of results on tabular data**: The goal of our work is to show that pointwise representation similarity measures such as PNKA are a useful and reliable tool to understand the effects of debiasing approaches. To establish the reliability of PNKA in this regard, we deliberately evaluate it on the well-understood COMPAS and Adult datasets and a popular debiasing approach. The fact that the insights we obtain with PNKA match the intuitive expectations and previous results in this domain shows that it indeed works as expected. Therefore, we can apply it confidently in the less studied domain of language embeddings, where we provide novel insights. We will highlight this motivation more clearly in the paper.
>
> 2. **Explaining the rationale for using relative positions of points**: We will expand the discussion in Section 2.1 to provide a clearer explanation for selecting relative position as a measure of similarity. Specifically, our choice aligns with widely accepted criteria for representation similarity metrics (RSMs) [1,2,3]. The CKA [1] paper discusses the ongoing debate in the community regarding the desirable properties of a robust RSM. We adopt the criteria outlined in that work, demonstrating theoretically in Appendix A.3 that PNKA satisfies these properties. We also empirically show in Appendix A.2 that its aggregate values align closely with those of established RSMs like CKA. Finally, regarding the intuition behind PNKA, the use of relative positioning for similarity assessment is well-established both in the representation similarity literature ([1,2,3]) and in neuroscience [4,5,6], where shifts in relative positioning are commonly used to analyze changes in brain representations.
>
> 3. **Scope of debiasing methods considered**: Our goal is to answer the question “What does PNKA reveal about the impact of a *specific* debiasing method on individual representations?”. Exhaustively comparing different debiasing approaches would answer a different question (“Which method works best in a given scenario?”) which is beyond our scope. If different methods yield different results, this reflects their distinct effects rather than providing a deeper insight into any one method. Instead, we show PNKA’s utility as an auditing tool across different domains, offering a structured way to assess how individual fairness methods modify representations. Expanding the study to cover multiple debiasing methods would significantly broaden the scope and potentially compromise the clarity of our findings. Instead, we believe a more meaningful validation lies in comparing PNKA’s insights with those from other metrics, which we already incorporate in our analysis.
>
> We hope this addresses your questions and are happy to provide further clarification if needed.
>
> References:
>
> [1] Kornblith, Simon, et al. "Similarity of neural network representations revisited." International conference on machine learning. PMLR, 2019.
>
> [2] Kriegeskorte, Nikolaus, Marieke Mur, and Peter A. Bandettini. "Representational similarity analysis-connecting the branches of systems neuroscience." Frontiers in systems neuroscience 2 (2008): 249.
>
> [3] Gower, John C. "Generalized procrustes analysis." Psychometrika 40 (1975): 33-51.
>
> [4] Kriegeskorte, Nikolaus, and Rogier A. Kievit. "Representational geometry: integrating cognition, computation, and the brain." Trends in cognitive sciences 17.8 (2013): 401-412.
>
> [5] Cichy, Radoslaw Martin, et al. "Dynamics of scene representations in the human brain revealed by magnetoencephalography and deep neural networks." NeuroImage 153 (2017): 346-358.
>
> [6] Mehrer, Johannes, et al. "An ecologically motivated image dataset for deep learning yields better models of human vision." Proceedings of the National Academy of Sciences 118.8 (2021): e2011417118.

---

### Author Response · Authors · 2025-02-05

We thank the reviewers for their time and thoughtful feedback. We have made the following revisions based on the feedback received:

**Clarity on the kernel used for PNKA:** In Section 2.2, we have explicitly stated that the linear kernel is used throughout the main paper. A footnote has been added mentioning that we also ran experiments with the RBF kernel, which are reported in the appendix. We have included RBF kernel experiments in Appendices A.1, B.1, C.1, and D, all of which yield results consistent with those obtained using the linear kernel.

**Justifying the choice of relative position of points for PNKA:** In Section 2.1, we have provided a justification for using the relative position of points to analyze representational change. To support this choice, we have incorporated citations from both the representation similarity metrics literature and neuroscience.

**Clarification of results on tabular data:** In Section 3, we have clarified that this section serves primarily as a case study to validate that PNKA is reliable and works as expected when auditing the effects of debiasing methods.

**Minor comments:** Figure captions for Figures 1 and 2 have been revised for improved clarity.

We hope these revisions address all concerns, but we are happy to make further improvements if needed.

---

### Decision · Action_Editor_ZffP · 2025-04-08

**Recommendation:** Accept as is

**Comment:**

Reviewers offered mixed recommendations, with some suggesting acceptance and others rejection. The main reason for rejection is the 'limited novelty and lack of a strong technical contribution.' However, all reviewers responded positively to the two acceptance criteria: claim and evidence, and audience. The authors have also provided additional analysis using the RBF kernel and found that the conclusions remain consistent. Please ensure the correct citation style is used for the camera-ready version.

**Audience:**

Yes.

**Claims And Evidence:**

The manuscript claims that Pointwise Normalized Kernel Alignment (PNKA) addresses two main limitations of existing debiasing measures such as equalized odds, equality of opportunity, and demographic parity: (1) current measures focus primarily on group- or aggregate-level analysis, and (2) they evaluate only model outputs on specific tasks, rather than task-agnostic representations. First, the manuscript presents experimental evidence on tabular data, showing the expected result that group fairness interventions often impact only a small subset of individuals while preserving representational similarity for the majority of the population. In contrast, individual fairness constraints lead to broader, more uniform changes across data points. The manuscript then turns to the less-studied domain of language embeddings, demonstrating that PNKA reveals how existing debiasing methods amplify gender information in gender-defining words, rather than effectively removing bias in gender-stereotypical words, as previously claimed.